# CLARIFY: Contrastive Preference Reinforcement Learning for Untangling Ambiguous Queries

Ni Mu [* 1]   Hao Hu [* 2]   Xiao Hu [1]   Yiqin Yang [3]   Bo Xu [3]   Qing-Shan Jia [1]

## Abstract

Preference-based reinforcement learning (PbRL) bypasses explicit reward engineering by inferring reward functions from human preference comparisons, enabling better alignment with human intentions. However, humans often struggle to label a clear preference between similar segments, reducing label efficiency and limiting PbRL's real-world applicability. To address this, we propose an offline PbRL method: **C**ontrastive **LeA**rning for **R**esolv**I**ng Ambiguous **F**eedback (CLARIFY), which learns a trajectory embedding space that incorporates preference information, ensuring clearly distinguished segments are spaced apart, thus facilitating the selection of more unambiguous queries. Extensive experiments demonstrate that CLARIFY outperforms baselines in both non-ideal teachers and real human feedback settings. Our approach not only selects more distinguished queries but also learns meaningful trajectory embeddings.

## 1. Introduction

Reinforcement Learning (RL) has achieved remarkable success in various domains, including robotics (Kalashnikov et al., 2018; Ju et al., 2022), gaming (Mnih et al., 2015; Ibarz et al., 2018), and autonomous systems (Schulman, 2015; Bellemare et al., 2020). However, a fundamental challenge in RL is the need for well-defined reward functions, which can be time-consuming and complex to design, especially when aligning them with human intent. To address this challenge, several approaches have emerged to learn

directly from human feedback or demonstrations, avoiding the need for explicit reward engineering. Preference-based Reinforcement Learning (PbRL) stands out by using human preferences between pairs of trajectory segments as the reward signal (Lee et al., 2021a; Christiano et al., 2017). This framework, based on pairwise comparisons, is easy to implement and captures human intent effectively.

However, when trajectory segments are highly similar, it becomes difficult for humans to differentiate between them and identify subtle differences (Inman, 2006; Bencze et al., 2021). Therefore, existing PbRL methods (Lee et al., 2021a; Liang et al., 2022; Park et al., 2022) face this significant challenge of **ambiguous queries**: humans struggle to give a clear signal for highly similar segments, leading to ambiguous preferences. As highlighted in previous work (Mu et al., 2024), this problem of ambiguous queries not only hinders labeling efficiency but also restricts the practical application of PbRL in real-world settings.

In this paper, we focus on solving this problem in offline PbRL by selecting a larger proportion of unambiguous queries. We propose a novel method, CLARIFY, which uses contrastive learning to learn trajectory embeddings and incorporate preference information, as outlined in Figure 1. CLARIFY features two contrastive losses to learn a meaningful embedding space. The ambiguity loss leverages the ambiguity information of queries, increasing the distance between embeddings of clearly distinguished segments while reducing the distance between ambiguous ones. The quadrilateral loss utilizes the preference information of queries, modeling the relationship between better and worse-performing segments as quadrilaterals. With theoretical guarantees, the two losses allow us to select more unambiguous queries based on the embedding space, thereby improving label efficiency.

Extensive experiments show the effectiveness CLARIFY. First, CLARIFY outperforms the state-of-the-art offline PbRL methods under non-ideal feedback from both scripted teachers and real human labelers. Second, CLARIFY significantly increases the proportion of unambiguous queries. We conduct human experiments to demonstrate that the queries selected by CLARIFY are more clearly distinguished, thereby improving human labeling efficiency. Fi-

*Equal contribution [1]Beijing Key Laboratory of Embodied Intelligence Systems, Department of Automation, Tsinghua University, Beijing, China [2]Moonshot AI, Beijing, China [3]The Key Laboratory of Cognition and Decision Intelligence for Complex Systems, Institute of Automation, Chinese Academy of Sciences, Beijing, China. Correspondence to: Yiqin Yang <yiqin.yang@ia.ac.cn>, Qing-Shan Jia <jiaqs@tsinghua.edu.cn>.

*Proceedings of the 42st International Conference on Machine Learning*, Vancouver, Canada. PMLR 267, 2025. Copyright 2025 by the author(s).

Figure 1: The framework of CLARIFY. 1) Train the trajectory embeddings via contrastive learning, incorporating preference information. 2) Using the embedding space, select clearly distinguished queries for non-ideal teachers via reject sampling.

nally, the visualization analysis of the learned embedding space reveals that segments with clearer distinctions are widely separated while similar segments are closely clustered together. This clustering behavior confirms the meaningfulness and coherence of the embedding space.

## 2. Related Work

**Preference-based RL (PbRL).** PbRL allows agents to learn from human preferences between pairs of trajectory segments, eliminating the need for reward engineering (Christiano et al., 2017). Previous PbRL works focus on improving feedback efficiency via query selection (Ibarz et al., 2018; Biyik et al., 2020), unsupervised pretraining (Lee et al., 2021a; Mu et al., 2024), and feedback augmentation (Park et al., 2022; Choi et al., 2024). Also, PbRL has been successfully applied to fine-tuning large language models (LLMs) (Ouyang et al., 2022), where human feedback serves as a more accessible alternative to reward functions.

In offline PbRL, an agent learns a policy from an offline trajectory dataset without reward signals, alongside preference feedback on segment pairs provided by a teacher. Traditional methods (Shin et al., 2023) follow a two-phase process: training a reward model from preference feedback, and then performing offline RL. Some works (Kim et al., 2022; Gao et al., 2024) focus on preference modeling, while others (Hejna & Sadigh, 2024; Kang et al., 2023) optimize policies directly from preference feedback.

**Learning from non-ideal feedback.** Despite PbRL's potential, real-world human feedback is often non-ideal, a growing area of concern. To address this, Lee et al. (2021b) proposes a model of non-ideal feedback. Cheng et al. (2024) assumes random label errors, identifying outliers via loss function values. Xue et al. (2023) improves reward robustness through regularization constraints. However, these approaches still rely on idealized models of feedback, which diverge from real human decision-making. Mu et al. (2024) is perhaps the closest work to ours, addressing the challenge of labeling similar segments in online PbRL, which aligns

with the ambiguous query issue. However, the method of Mu et al. (2024) cannot be applied to offline settings. This paper focuses on addressing this issue in the offline PbRL.

**Contrastive learning for RL.** Contrastive learning is widely used in self-supervised learning to differentiate between positive and negative samples and extract meaningful features (Oord et al., 2018; Chen & He, 2021). In RL, it has been applied to representation learning in image-based RL (Laskin et al., 2020; Yuan & Lu, 2022) to improve learning efficiency, and to temporal distance learning (Myers et al., 2024; Jiang et al., 2024) to encourage exploration. In this paper, we apply contrastive learning to incorporate preference into trajectory representations for offline PbRL.

## 3. Preliminaries

**Preference-based RL.** In RL, an agent aims to maximize the cumulative discounted rewards in a Markov Decision Process (MDP), defined by a tuple $(S, A, P, r, \gamma)$. $S$ and $A$ are the state and action space, $P = P(\cdot|s, a)$ is the environment transition dynamics, $r = r(s, a)$ is the reward function, and $\gamma$ is the discount factor. In offline PbRL, the true reward function $r$ is unknown, and we have an offline dataset $D$ without reward signals. We request preference feedback $p$ for two trajectory segments $\sigma_0$ and $\sigma_1$ of length $H$ sampled from $D$. $p = 0$ denotes $\sigma_1$ is preferred ($\sigma_1 \succ \sigma_0$), $p = 1$ denotes the opposite ($\sigma_1 \prec \sigma_0$). In cases where segments are too similar to distinguish, we set $p = \texttt{no\_cop}$, and this label is skipped during learning. The preference data $(\sigma_0, \sigma_1, p)$ forms the preference dataset $D_p$.

Conventional offline PbRL methods estimate a reward function $\hat{r} = \hat{r}_\psi(s, a)$ from $D_p$ and use it to train a policy $\pi_\theta$ via offline RL. The Preference model, typically based on the Bradley-Terry model (Bradley & Terry, 1952), estimates the probability $P_\psi[\sigma_1 \succ \sigma_0]$ as:

$$P_\psi[\sigma_1 \succ \sigma_0] = \frac{\exp \sum_t \hat{r}_\psi(s_t^1, a_t^1)}{\sum_{i \in \{0,1\}} \exp \sum_t \hat{r}_\psi(s_t^i, a_t^i)}. \quad (1)$$

The following cross-entropy loss is minimized:

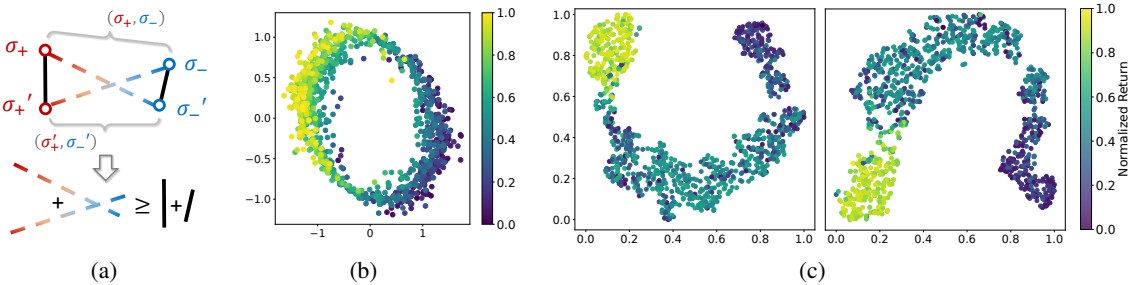

Figure 2: Illustration of quadrilateral loss $\mathcal{L}_{\text{quad}}$. (a) A demonstration of the main idea of $\mathcal{L}_{\text{quad}}$. (b) An embedding visualization of the intuitive example in Section 4.1. (c) True embedding visualization results of the benchmark experiments on the "drawer-open" task, under skip rate $\epsilon = 0.5$ and $\epsilon = 0.7$.

$$\min_{\psi} \mathcal{L}_{\text{reward}} = - \mathbb{E}_{(\sigma_0, \sigma_1, p) \sim D_p} \Big[ (1-p) \log P_{\psi}[\sigma_0 \succ \sigma_1] + p \log P_{\psi}[\sigma_1 \succ \sigma_0] \Big]. \tag{2}$$

**Contrastive learning.** Contrastive learning is a self-supervised learning approach, which learns meaningful representations by comparing pairs of samples. The basic idea is to bring similar (positive) samples closer in the embedding space while pushing dissimilar (negative) samples apart. Early works (Chopra et al., 2005) directly optimize this objective, using the following loss function:

$$\mathcal{L}_{\text{contrastive}} = \frac{1}{2N} \sum_{i=1}^{N} \Big[ \mathbb{I}_{(y_i = y_j)} \|z_i - z_j\|_2^2 + \mathbb{I}_{(y_i \neq y_j)} \max(0, m - \|z_i - z_j\|_2)^2 \Big], \tag{3}$$

where $z_i$ is the representation of sample $i$, $\mathbb{I}_{y_i = y_j}$ and $\mathbb{I}_{(y_i \neq y_j)}$ are indicator functions, and $m$ is a pre-defined margin. InfoNCE loss (Oord et al., 2018) is widely used in recent works, which is formulated as:

$$\mathcal{L}_{\text{InfoNCE}} = - \log \frac{\exp(\text{sim}(z_i, z_i')/t)}{\sum_{k=1}^{K} \exp(\text{sim}(z_i, z_k)/t)}, \tag{4}$$

where $\text{sim}(\cdot, \cdot)$ is the similarity function, $z_i'$ denotes the positive sample of $z_i$, and $t$ is a temperature scaling parameter. In our work, we apply contrastive learning to model trajectory representations, incorporating preference information. The encoder $z = f_\phi(\tau)$ models arbitrary-length trajectory $\tau$, and its corresponding preference-based loss functions are detailed in Section 4.1.

## 4. Method

In this section, we start by introducing the issue of **ambiguous queries**: humans struggle to clearly distinguish between similar trajectory segments, making the query ambiguous. This issue, validated in human experiments (Mu et al., 2024), hinders the practical application of PbRL. While previous work (Mu et al., 2024) tackles this issue in online PbRL settings, it remains unsolved in offline settings.

To tackle ambiguous queries in offline PbRL, we propose a novel method, **C**ontrastive **LeA**rning for **R**esolv**I**ng Ambiguous **F**eedback (CLARIFY), which maximizes the selection of clearly-distinguished queries to improve human labeling efficiency. Our approach is based on contrastive learning, integrating preference information into trajectory embeddings. In the learned embedding space, clearly distinguished segments are well-separated, while ambiguous segments remain close, as detailed in Section 4.1. Based on this embedding, we introduce a query selection method in Section 4.2 to select more unambiguous queries. The overall framework of CLARIFY is illustrated in Figure 1 and Algorithm 1, with implementation details provided in Section 4.3.

### 4.1. Representation Learning

In this subsection, we first formalize the problem of preference-based representation learning. Our goal is to train an encoder $z = f_\phi(\tau)$, where $z$ is a fixed-dimensional embedding of trajectory $\tau$. We leverage a preference dataset $D_p$ containing tuples $(\sigma_0, \sigma_1, p)$, where $\sigma_0$ and $\sigma_1$ are trajectory segments, and $p \in \{0, 1, \texttt{no\_cop}\}$ indicates the preference: $p = 0$ if $\sigma_0$ is preferred, $p = 1$ if $\sigma_1$ is preferred, and $p = \texttt{no\_cop}$ if the segments are too similar to compare.

A well-structured embedding space should maximize the distance between clearly distinguished segments while minimizing the distance between ambiguous ones. In such a space, trajectories with similar performance are close, while those with large performance gaps are far apart. This results in a meaningful and coherent embedding space, where high-performance trajectories form one cluster, low-

performance trajectories form another, and intermediate trajectories smoothly transition between them. To achieve this, we propose two contrastive learning losses.

**Ambiguity loss $\mathcal{L}_{\text{amb}}$.** The first loss, called the ambiguity loss, directly optimizes this goal. It maximizes the distance between the embeddings of clearly distinguished segment pairs and minimizes the distance between ambiguous ones:

$$\min_\phi \mathcal{L}_{\text{amb}} = \Big[ - \mathop{\mathbb{E}}_{\substack{(\sigma_0,\sigma_1,p) \sim D_p, \\ p \in \{0,1\}}} \ell(z_0, z_1) \\ + \mathop{\mathbb{E}}_{\substack{(\sigma_0,\sigma_1,p) \sim D_p, \\ p=\text{no\_cop}}} \ell(z_0, z_1) \Big], \quad (5)$$

where $z_0 = f_\phi(\sigma_0)$, $z_1 = f_\phi(\sigma_1)$, and $\ell(\cdot, \cdot)$ is the distance metric.

However, relying solely on the ambiguity loss $\mathcal{L}_{\text{amb}}$ can cause several issues. First, when the preference dataset is small, continuous optimization of $\mathcal{L}_{\text{amb}}$ may lead to overfitting. Additionally, $\mathcal{L}_{\text{amb}}$ alone can cause representation collapse, where ambiguous segments are mapped to the same point in the embedding space. This is because $\mathcal{L}_{\text{amb}}$ only leverages ambiguity information (whether segments are distinguishable) but ignores preference relations (which segment is better). The quadrilateral loss $\mathcal{L}_{\text{quad}}$, introduced below, mitigates this issue by acting as a regularizer, ensuring the embedding space better captures the full preference structure and improving representation quality.

**Quadrilateral loss $\mathcal{L}_{\text{quad}}$.** To address these issues, we introduce the quadrilateral loss $\mathcal{L}_{\text{quad}}$, which directly models preference relations between segment pairs, better capturing the underlying structure in the embedding space. Specifically, for two clearly distinguished queries $(\sigma_+, \sigma_-)$ and $(\sigma'_+, \sigma'_-)$, where $\sigma_+$ and $\sigma'_+$ are preferred over $\sigma_-$ and $\sigma'_-$, we create a quadrilateral-shaped relationship between their embeddings. By utilizing pairs of queries, the training data size grows from $O(n)$ to $O(n^2)$, mitigating the overfitting issue caused by limited preference data.

The core idea $\mathcal{L}_{\text{quad}}$ is illustrated in Figure 2(a). For each pair of clearly distinguished queries, we treat $\sigma_+$ and $\sigma'_+$ as "positive" samples, as they are preferred over $\sigma_-$ and $\sigma'_-$, the "negative" samples. The quadrilateral loss encourages the sum of distances between positive samples $\sigma_+, \sigma'_+$ and between negative samples $\sigma_-, \sigma'_-$ to be smaller than the sum of distances within positive or negative pairs (i.e., between $\sigma_+$ and $\sigma_-$, or $\sigma'_+$ and $\sigma_-$). Formally, this is expressed as:

$$\min_\phi \mathcal{L}_{\text{quad}} = - \mathop{\mathbb{E}}_{((\sigma_+,\sigma_-),(\sigma'_+,\sigma'_-)) \sim D_p} \Big[ \ell(z^+, z^{-'}) \\ + \ell(z^{+'}, z^-) - \ell(z^+, z^{+'}) - \ell(z^-, z^{-'}) \Big], \quad (6)$$

where $z^+ = f_\phi(\sigma_+)$, $z^{+'} = f_\phi(\sigma'_+)$, $z^- = f_\phi(\sigma_-)$, $z^{-'} = f_\phi(\sigma'_-)$.

**An intuitive example.** To demonstrate the effectiveness of the quadrilateral loss, we conducted a simple experiment. We generated 1000 data points with values uniformly distributed between 0 and 1, initializing each data point's embedding as a 2-dim vector sampled from a standard normal distribution. We then optimized these embeddings using the quadrilateral loss. As shown in Figure 2(b), the learned embedding space exhibits a smooth and meaningful distribution, with higher-valued data points transitioning gradually to lower-valued ones. Please refer to Appendix C.3 for experimental details.

### 4.2. Query Selection

This section presents a query selection method based on rejection sampling to select more unambiguous queries. For a given query $(\sigma_0, \sigma_1, p)$, we calculate the embedding distance $d_{\text{emb}} = \ell(f_\phi(\sigma_0), f_\phi(\sigma_1))$ between its two segments. For a batch of queries, we can obtain a distribution $p(d_{\text{emb}})$ of the embedding distances. We aim to manipulate this distribution using rejection sampling to increase the proportion of clearly distinguished queries.

We define the rejection sampling distribution $q(d_{\text{emb}})$ as follows. First, we estimate the density functions $\rho_{\text{clr}}(d_{\text{emb}})$ and $\rho_{\text{amb}}(d_{\text{emb}}) = 1 - \rho_{\text{clr}}(d_{\text{emb}})$ for clearly-distinguished and ambiguous pairs, using the existing preference dataset $D_p$. These densities reflect the likelihood of observing a distance $d_{\text{emb}}$ for each type of segment pair. Next, we compute a new density function:

$$\rho_1(d_{\text{emb}}) = \frac{\max(0, \rho_{\text{clr}}(d_{\text{emb}}) - \rho_{\text{amb}}(d_{\text{emb}}))}{\int \max(0, \rho_{\text{clr}}(d') - \rho_{\text{amb}}(d')) \, dd'}, \\ \rho_2(d_{\text{emb}}) = \frac{\rho_{\text{clr}}(d_{\text{emb}})/\rho_{\text{amb}}(d_{\text{emb}})}{\int (\rho_{\text{clr}}(d')/\rho_{\text{amb}}(d')) \, dd'}, \quad (7) \\ \rho(d_{\text{emb}}) = 0.5(\rho_1(d_{\text{emb}}) + \rho_2(d_{\text{emb}})).$$

This density function emphasizes the distances where clearly distinguished segments are more frequent than ambiguous ones. Finally, we multiply this new density function by the original distribution $p(d_{\text{emb}})$ to obtain the rejection sampling distribution $q(d_{\text{emb}})$:

$$q(d_{\text{emb}}) = p(d_{\text{emb}}) \cdot \rho(d_{\text{emb}}), \quad (8)$$

which increases the likelihood of selecting queries with clearly distinguished segment pairs, improving labeling efficiency. For the remaining queries after rejection sampling, we follow prior work (Lee et al., 2021a; Shin et al., 2023) by selecting those with maximum disagreement.

### 4.3. Implementation Details

**Embedding training.** For embedding training in Section 4.1, we adopt a Bi-directional Decision Transformer (BDT) architecture (Furuta et al., 2021a), where the encoder $z =$

Table 1: Success rates on Metaworld tasks (the first 7 tasks) and episodic returns on DMControl tasks (the last 2 tasks), over 6 random seeds. We use skip rate $\epsilon = 0.5, 0.7$ and report the average performance and standard deviation of the last 5 trained policies. The yellow and gray shading represent the best and second-best performances, respectively.

| Skip Rate | Algorithm | box-close | dial-turn | drawer-open | handle -pull-side | hammer | peg-insert -side | sweep-into | cheetah-run | walker-walk |
|---|---|---|---|---|---|---|---|---|---|---|
| - | IQL | 94.90 ± 1.42 | 76.50 ± 1.73 | 98.60 ± 0.45 | 99.30 ± 0.59 | 72.00 ± 2.35 | 88.40 ± 1.30 | 79.40 ± 1.91 | 607.46 ± 8.13 | 830.15 ± 20.08 |
| 0.5 | MR | 0.26 ± 0.02 | 14.46 ± 5.27 | 50.47 ± 6.24 | 79.73 ± 12.19 | 0.14 ± 0.08 | 9.23 ± 2.03 | 23.25 ± 9.67 | 205.04 ± 50.53 | 322.93 ± 161.07 |
| | OPRL | **8.25** ± 3.16 | **57.33** ± 25.02 | **72.67** ± 2.87 | 83.92 ± 7.98 | 14.80 ± 5.27 | **22.00** ± 4.64 | **61.00** ± 7.52 | 531.96 ± 48.75 | 646.40 ± 51.35 |
| | PT | 0.15 ± 0.20 | 30.54 ± 7.24 | 61.64 ± 10.07 | **89.75** ± 6.07 | 0.11 ± 0.10 | 10.20 ± 2.94 | 46.35 ± 3.35 | 384.59 ± 99.26 | 599.43 ± 39.42 |
| | OPPO | 0.56 ± 0.79 | 13.06 ± 12.74 | 11.67 ± 6.24 | 0.56 ± 0.79 | 2.78 ± 4.78 | 0.00 ± 0.00 | 15.56 ± 11.57 | 346.01 ± 127.78 | 311.27 ± 42.73 |
| | LiRE | 3.60 ± 0.69 | 46.20 ± 6.75 | 63.47 ± 10.38 | 52.07 ± 33.58 | **16.20** ± 13.37 | 21.60 ± 4.00 | 57.47 ± 2.74 | **553.61** ± 43.16 | **789.18** ± 28.77 |
| | CLARIFY | **29.40** ± 16.27 | **77.50** ± 7.37 | **83.50** ± 7.40 | **95.00** ± 1.22 | **26.75** ± 11.26 | **24.25** ± 6.65 | **68.00** ± 7.13 | **617.31** ± 14.43 | **796.34** ± 12.87 |
| 0.7 | MR | 0.20 ± 0.11 | 16.57 ± 8.22 | 45.51 ± 10.25 | 74.82 ± 17.10 | 0.06 ± 0.09 | 5.21 ± 2.63 | 17.22 ± 6.05 | 234.77 ± 81.40 | 306.39 ± 134.72 |
| | OPRL | 7.40 ± 6.97 | **63.40** ± 9.46 | 46.50 ± 18.83 | **84.00** ± 7.11 | 5.00 ± 2.28 | **23.75** ± 5.36 | 52.00 ± 9.33 | 513.94 ± 55.45 | 664.16 ± 89.47 |
| | PT | 0.18 ± 0.18 | 25.64 ± 7.40 | **64.72** ± 18.44 | 74.94 ± 12.56 | 0.09 ± 0.05 | 8.21 ± 4.07 | 28.52 ± 6.67 | 429.19 ± 44.92 | 647.68 ± 38.53 |
| | OPPO | 0.56 ± 0.79 | 2.78 ± 2.83 | 11.67 ± 6.24 | 0.56 ± 0.79 | 4.44 ± 6.29 | 0.00 ± 0.00 | 15.56 ± 11.57 | 355.69 ± 59.35 | 304.19 ± 16.25 |
| | LiRE | **7.67** ± 16.79 | 38.40 ± 8.52 | 39.10 ± 6.78 | 50.60 ± 31.59 | **14.25** ± 7.30 | 23.40 ± 3.40 | 56.00 ± 6.40 | **514.75** ± 10.02 | 795.02 ± 22.80 |
| | CLARIFY | **17.50** ± 6.87 | **79.40** ± 3.83 | **78.60** ± 10.52 | **95.00** ± 1.10 | **28.75** ± 15.58 | 23.00 ± 1.22 | **59.67** ± 10.09 | **593.24** ± 22.75 | **816.54** ± 11.08 |

$f_\phi(\tau)$ and the decoder $\hat{a} = \pi_{\phi'}(s, z)$ are Transformer-based. The model is trained with a reconstruction loss:

$$\min_{\phi,\phi'} \mathcal{L}_{\text{recon}} = \mathbb{E}_{\substack{\tau \sim D \\ (s,a) \sim \tau}} \left\| \pi_{\phi'}\left(s, f_\phi(\tau)\right), a \right\|_2. \qquad (9)$$

Appendix E provides more details on BDT. Also, to stabilize training, we constrain the L2 norm of the embeddings to be close to 1. Without this constraint, embeddings may either grow unbounded or collapse to the origin, both of which can cause training to fail:

$$\min_\phi \mathcal{L}_{\text{norm}} = \mathbb{E}_{\tau \sim D} \max\left(\|f_\phi(\tau)\|_2, 1\right). \qquad (10)$$

The final loss function combines these terms:

$$\mathcal{L} = \mathcal{L}_{\text{recon}} + \lambda_{\text{amb}} \mathcal{L}_{\text{amb}} + \lambda_{\text{quad}} \mathcal{L}_{\text{quad}} + \lambda_{\text{norm}} \mathcal{L}_{\text{norm}}, \quad (11)$$

where $\lambda_{\text{amb}}, \lambda_{\text{quad}}, \lambda_{\text{norm}}$ are hyperparameters. We use the L2 distance as the distance metric $\ell(\cdot, \cdot)$.

**Rejection sampling.** For rejection sampling in Section 4.2, following the successful discretization of low-dimensional features in prior RL works (Bellemare et al., 2017; Furuta et al., 2021b), we discretize the embedding distance $d_{\text{emb}}$ it into $n_{\text{bin}}$ intervals to handle continuous distributions.

Based on the above discussion, the overall process of CLARIFY is summarized in Algorithm 1. First, we randomly sample a query batch to pretrain the encoder and reward model. Next, we select queries based on the pretrained embedding space, update the preference dataset $D_p$ and reward model, and retrain the embedding using the updated $D_p$. Finally, we train the policy $\pi_\theta$ using a standard offline RL algorithm, such as IQL (Kostrikov et al., 2021). Additional implementation details are provided in Appendix C.2.

## 5. Theoretical Analysis

This section establishes the theoretical foundation of CLARIFY's embedding framework and provides insights into the proposed objective. We present two key propositions showing that the losses $\mathcal{L}_{\text{quad}}$ and $\mathcal{L}_{\text{amb}}$ ensure: 1) margin separation for distinguishable samples, and 2) convex separability of preference signals in the embedding space. The first property guarantees a meaningful geometric embedding for selecting distinguishable samples, while the second ensures that the embedding space does not collapse and maintains a clear separation between positive and negative samples.

**Margin guarantees for disentangled representations.** Proposition 5.1 formalizes how $\mathcal{L}_{\text{amb}}$ enforces geometric separation between trajectories with distinguishable value differences.

**Proposition 5.1** (Positive Separation Margin under Optimal Ambiguity Loss). *Let $D_p$ be a distribution over labeled trajectory pairs $(\sigma_0, \sigma_1, p)$ with $p \in \{0, 1, \texttt{no\_cop}\}$. Define $P = \{(\sigma_0, \sigma_1) \mid p \in \{0, 1\}\}$ and $N = \{(\sigma_0, \sigma_1) \mid p = \texttt{no\_cop}\}$. Let*

$$\phi^* = \arg\min_\phi \mathcal{L}_{\text{amb}}(\phi), \qquad z_i = f_{\phi^*}(\sigma_i),$$

*and set*

$$d_{\min}^+ = \inf_{(\sigma_0, \sigma_1) \in P} \ell(z_0, z_1),$$

$$d_{\max}^- = \sup_{(\sigma_0, \sigma_1) \in N} \ell(z_0, z_1),$$

*where $\ell(z_0, z_1) = \|z_0 - z_1\|$. If the class $\{f_\phi\}$ is continuous and sufficiently expressive, and $\text{supp}(D_p)$ is compact in $(\sigma_0, \sigma_1)$-space, then there exists $\delta > 0$ such that*

$$d_{\min}^+ \geq d_{\max}^- + \delta > 0.$$

*Proof.* Please refer to Appendix A.1 for detailed proof. □

This proposition guarantees that the ambiguity-loss minimizer $\phi^*$ yields not only a correct ordering of pairwise distances but also enforces a strict margin $\delta > 0$:

$$\inf_{(\sigma_0, \sigma_1) \in P} \|z_0 - z_1\| \geq \sup_{(\sigma_0, \sigma_1) \in N} \|z_0 - z_1\| + \delta.$$

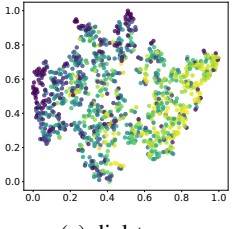
(a) dial-turn

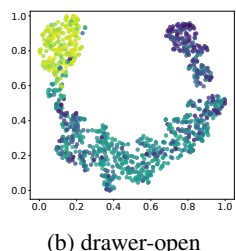
(b) drawer-open

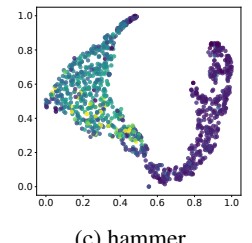
(c) hammer

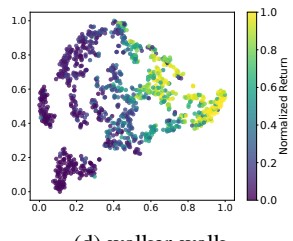
(d) walker-walk

Figure 3: Visualizations of the learned embedding spaces under $\epsilon = 0.5$, where segments with high returns (bright color) and low returns (dark color) are clearly separated into distinct clusters, with a smooth transition between their centers.

In effect, CLARIFY's objective $\mathcal{L}_{\text{amb}}$ provably carves out a uniform buffer $\delta$ around all distinguishable trajectory pairs, endowing the learned embedding space with a nontrivial geometric margin that enhances robustness and separability.

**Convex geometry of preference signals.** The second proposition characterizes how $\mathcal{L}_{\text{quad}}$ induces convex separability in the embedding space. We show that minimizing this loss directly translates to constructing a robust decision boundary between preferred and non-preferred trajectories:

**Proposition 5.2** (Convex Separability). *Assume the positive and negative samples are distributed in two convex sets $\mathcal{C}^+$ and $\mathcal{C}^-$ in the embedding space. Let $\mu^+ = \mathbb{E}[z^+]$ and $\mu^- = \mathbb{E}[z^-]$ denote the class centroids. If $\mathcal{L}_{\text{quad}}$ is minimized with a margin $\eta > 0$, then there exists a hyperplane $\mathcal{H}$ defined by:*

$$\mathcal{H} = \{z \in \mathbb{R}^d \mid w^T z + b = 0\}$$

*such that for all $z^+ \in \mathcal{C}^+$ and $z^- \in \mathcal{C}^-$,*

$$\tilde{d}(z^+, \mathcal{H}) \geq \eta \quad and \quad \tilde{d}(z^-, \mathcal{H}) \leq -\eta$$

*where $\tilde{d}(z, \mathcal{H}) = (w^\top b + z)/||w||$ denotes the signed distance from $z$ to $\mathcal{H}$.*

*Proof.* Please refer to Appendix A.2 for detailed proof. □

The second proposition characterizes the separability achieved in the embedding space. The centroid separation $\mu^+ - \mu^-$ acts as a global discriminator, while the margin $\eta$ ensures local robustness against ambiguous samples near decision boundaries. This result has two key implications: 1) The existence of a separating hyperplane $\mathcal{H}$ guarantees linear separability of preferences, enabling simple query selection policies (e.g., margin-based sampling) to achieve high labeling efficiency. 2) The margin $\eta$ directly quantifies the "safety gap" against ambiguous queries, where any query pair within $2\eta$ distance would be automatically filtered out as unreliable. This mathematically substantiates our method's ability to *actively avoid* ambiguous queries during human feedback collection.

To better illustrate the dynamics of the embedding space of the proposed losses, we provide a gradient-based analy-

sis to explain, available in Appendix A.3. Overall, CLARIFY's framework combines three principles: 1) Margin maximization for query disambiguation, 2) Convex geometric separation for reliable hyperplane decisions, 3) Dynamically balanced contrastive gradients for stable embedding learning. Together, these properties ensure that the learned representation is both *geometrically meaningful* (aligned with trajectory values) and *algorithmically useful* (enabling efficient query selection).

# 6. Experiments

We designed our experiments to answer the following questions: *Q1:* How does CLARIFY compare to other state-of-the-art methods under non-ideal teachers? *Q2:* Can CLARIFY improve label efficiency by query selection? *Q3:* Can CLARIFY learn a meaningful embedding space for trajectory representation? *Q4:* What is the contribution of each of the proposed techniques in CLARIFY?

## 6.1. Setups

**Dataset and tasks.** Previous offline PbRL studies often use D4RL (Fu et al., 2020) for evaluation, but D4RL is shown to be insensitive to reward learning due to the "survival instinct" (Li et al., 2023), where performance can remain high even with wrong rewards (Shin et al., 2023). To address this, we use the offline dataset presented by Choi et al. (2024) with Metaworld (Yu et al., 2020) and DMControl (Tassa et al., 2018), which has been proven to be suitable for reward learning (Choi et al., 2024). Specifically, we choose 7 complex Metaworld tasks: box-close, dial-turn, drawer-open, handle-pull-side, hammer, peg-insert-side, sweep-into, and 2 complex DMControl tasks: cheetah-run, walker-walk. Task details are provided in Appendix C.1.

**Baselines.** We compare CLARIFY with several state-of-the-art methods, including Markovian Reward (MR), Preference Transformer (PT) (Kim et al., 2022), OPRL (Shin et al., 2023), OPPO (Kang et al., 2023), and LiRE (Choi et al., 2024). MR is based on a Markovian reward model using MLP layers, serving as the baseline in PT, which uses a Transformer for reward modeling. OPRL leverages reward

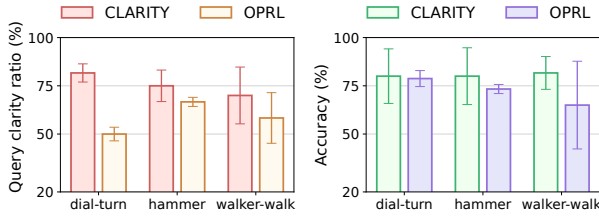

Figure 4: Query clarity ratio and accuracy of human labels for CLARIFY and OPRL, with CLARIFY-selected queries are more clearly distinguished for humans.

ensembles and selects queries with maximum disagreement. OPPO also learns trajectory embeddings but optimizes the policy directly in the embedding space. LiRE uses the listwise comparison to augment the feedback. Following prior work, we use IQL (Kostrikov et al., 2021) to optimize the policy after reward learning. Also, we train IQL with ground truth rewards as a performance upper bound. More implementation details are provided in Appendix C.2.

**Non-ideal teacher design.** Following prior works (Lee et al., 2021a; Shin et al., 2023), we use a scripted teacher for systematic evaluation, which provides preferences between segments based on the sum of ground truth rewards $r_{\mathrm{gt}}$. To better mimic human decision-making uncertainty, we introduce a "skip" mechanism. When the performance difference between two segments $\sigma_0, \sigma_1$ is marginal, that is,

$$\left| \sum_{(s,a)\in\sigma_0} r_{\mathrm{gt}}(s,a) - \Sigma_{(s,a)\in\sigma_1} r_{\mathrm{gt}}(s,a) \right| < \epsilon H \cdot r_{\mathrm{avg}}, \quad (12)$$

the teacher skips the query by assigning $p = \texttt{no\_cop}$. Here, $H$ is the segment length, and $r_{\mathrm{avg}}$ is the average ground truth reward for transitions in offline dataset $D$. We refer to $\epsilon \in (0,1)$ as the skip rate. This model is similar to the "threshold" mechanism in Choi et al. (2024), but differs from the "skip" teacher in B-Pref (Lee et al., 2021b), which skips segments with too small returns.

## 6.2. Evaluation on the Offline PbRL Benchmark

**Benchmark results.** We compare CLARIFY with baselines on Metaworld and DMControl. Table 1 shows that skipping degrades MR's performance, even with PT. LiRE improves MR via listwise comparison, while OPRL enhances performance by selecting maximally disagreed queries. OPPO is similarly affected by skipping, performing on par with MR. In contrast, CLARIFY selects clearer queries, improving reward learning and achieving the best results in most tasks.

**Enhanced query clarity.** To assess CLARIFY's ability to select unambiguous queries, we compare the distinguishable query ratio under a non-ideal teacher. Table 3 shows CLARIFY achieves higher query clarity across tasks.

We further validate this via human experiments on dial-

Table 2: Performance of CLARIFY and MR using different numbers of queries, under skip rate $\epsilon = 0.5$.

| # of Queries | dial-turn | | sweep-into | |
|---|---|---|---|---|
| | CLARIFY | MR | CLARIFY | MR |
| 100 | $59.50 \pm 4.67$ | $49.50 \pm 10.16$ | $54.00 \pm 2.16$ | $49.67 \pm 4.03$ |
| 500 | $77.25 \pm 6.87$ | $50.50 \pm 5.98$ | $68.67 \pm 1.70$ | $56.67 \pm 4.03$ |
| 1000 | $77.50 \pm 3.01$ | $57.33 \pm 5.02$ | $68.00 \pm 3.19$ | $61.00 \pm 7.52$ |
| 2000 | $77.80 \pm 6.10$ | $59.00 \pm 5.72$ | $68.75 \pm 1.48$ | $63.25 \pm 6.02$ |

Table 3: Ratios of clearly-distinguished queries under the non-ideal teacher with $\epsilon = 0.5$, for CLARIFY and baselines.

| | dial-turn | hammer | sweep-into | walker-walk |
|---|---|---|---|---|
| CLARIFY | **76.33%** | **62.67%** | **68.67%** | **46.86%** |
| MR | 46.95% | 50.33% | 36.67% | 36.28% |
| OPRL | 31.67% | 12.67% | 25.60% | 15.64% |
| PT | 43.90% | 49.33% | 31.67% | 37.90% |

turn, hammer, and walker-walk. Labelers provide 20 preference labels per run over 3 seeds, selecting the segment best achieving the task (e.g., upright posture in walker-walk). Unclear queries are skipped. Appendix D details task objectives and prompts.

We evaluate with:

1) Query clarity: the ratio of queries with human-provided preferences.
2) Accuracy: agreement between human labels and ground truth.

Figure 4 confirms CLARIFY's superior query clarity and accuracy, validating its effectiveness.

**Embedding space visualizations.** We visualize the learned embedding space using t-SNE in Figure 3. Each point represents a segment embedding, colored by its normalized return value. For most tasks, high-performance segments (bright colors) and low-performance segments (dark colors) form distinct clusters with smooth transitions, indicating that the embedding space effectively captures performance differences. The balanced distribution of points further demonstrates the stability and quality of the learned representation, highlighting CLARIFY's ability to create a meaningful and coherent embedding space.

## 6.3. Human Experiments

**Validation of the non-ideal teacher.** To validate the appropriateness of our non-ideal teacher (Section 6.1), we conduct a human labeling experiment, where labelers provide preferences between segment pairs with varying return differences. The results in Figure 5 show that as the return difference increases, both the query clarity ratio and accuracy improve. This suggests that when the return difference is small, humans struggle to distinguish between segments,

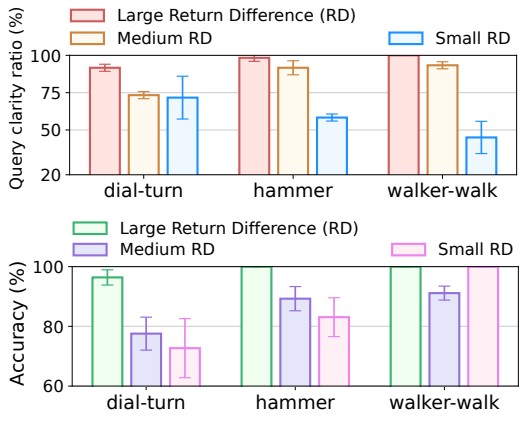

Figure 5: Query clarity ratio and accuracy of human labels for queries with varying return differences (RD). For dial-turn and hammer, RD values are 300, 100, and 10; for walker-walk, RD are 30, 10, and 1.

Table 4: Performance of CLARIFY and $\rho_{clr}$-based query selection method, under skip rate $\epsilon = 0.5$.

| Query Selection | dial-turn | sweep-into |
|---|---|---|
| CLARIFY | **77.50** $\pm$ 3.01 | **68.00** $\pm$ 3.19 |
| $\rho_{clr}$-based | 61.33 $\pm$ 5.31 | 48.60 $\pm$ 10.74 |

aligning with our assumption that small return differences lead to ambiguous queries.

**Human evaluation.** To evaluate CLARIFY's performance with real human preferences, we conduct experiments comparing CLARIFY with OPRL on the walker-walk task across 3 random seeds. Human labelers provide 100 feedback samples per run, with preference batch size $M = 20$. As shown in Table 5, CLARIFY outperforms OPRL in policy performance, suggesting that our reward models have higher quality. Additionally, queries selected by CLARIFY have higher clarity ratios and accuracy in human labeling, indicating that our approach improves the preference labeling process by selecting more clearly distinguished queries. For more details on the human experiments, please refer to Appendix D. We believe these results indicate CLARIFY's potential in real-world applications, especially those involving human feedback, such as LLM alignment.

### 6.4. Ablation Study

**Component analysis.** To assess the impact of each contrastive loss in CLARIFY, we incrementally apply the ambiguity loss $\mathcal{L}_{amb}$ and the quadrilateral loss $\mathcal{L}_{quad}$. As shown in Table 6, without either loss, performance is similar to OPRL. Using only $\mathcal{L}_{amb}$ yields unstable results due to overfitting early in training. When only $\mathcal{L}_{quad}$ is applied, performance improves but with slower convergence. When both losses are used, the best results are achieved, showing that their

Table 5: Performance of CLARIFY and OPRL on walker-walk task, under real human labelers.

| | CLARIFY | OPRL |
|---|---|---|
| Episodic Returns | **420.75** $\pm$ 52.02 | 265.91 $\pm$ 33.57 |
| Query Clarity Ratio (%) | **63.33** $\pm$ 8.50 | 53.33 $\pm$ 6.24 |
| Accuracy (%) | **87.08** $\pm$ 9.15 | 66.67 $\pm$ 4.71 |

Table 6: Performance of CLARIFY with and without optimizing $\mathcal{L}_{amb}$ and $\mathcal{L}_{quad}$, under skip rate $\epsilon = 0.5$.

| $\mathcal{L}_{amb}$ | $\mathcal{L}_{quad}$ | dial-turn | sweep-into |
|---|---|---|---|
| ✗ | ✗ | 63.20 $\pm$ 4.79 | 40.00 $\pm$ 11.29 |
| ✓ | ✗ | 69.00 $\pm$ 11.20 | 52.80 $\pm$ 17.01 |
| ✗ | ✓ | 71.25 $\pm$ 8.81 | 62.20 $\pm$ 4.92 |
| ✓ | ✓ | 77.50 $\pm$ 3.01 | 68.00 $\pm$ 3.19 |

combination is crucial to the method's success.

**Ablation on the query selection.** We compare two query selection methods: CLARIFY 's rejection sampling and a density-based approach that selects the highest-density queries $\rho_{clr}(d)$, aiming for the most clearly distinguished queries. As shown in Table 4, the density-based method performs poorly, likely due to selecting overly similar queries, reducing diversity in the queries. In contrast, CLARIFY selects a more diverse set of unambiguous queries, yielding better performance.

**Enhanced query efficiency.** We compare the performance of CLARIFY and MR using different numbers of queries. As shown in Table 2, CLARIFY outperforms MR consistently, even if only 100 queries are provided. The result demonstrates CLARIFY's ability to make better use of the limited feedback budget.

**Ablation on hyperparameters.** As Figure 6 visualizes, when varying the values of $\lambda_{dist}$, $\lambda_{quad}$, and $\lambda_{norm}$, the quality of embedding space remains largely unaffected. This demonstrates the robustness of CLARIFY to hyperparameter changes, as small adjustments do not significantly alter the quality of learned embeddings.

## 7. Conclusion

This paper presents CLARIFY, a novel approach that enhances PbRL by addressing the **ambiguous query** issue. Leveraging contrastive learning, CLARIFY learns trajectory embeddings with preference information and employs reject sampling to select more clearly distinguished queries, improving label efficiency. Experiments show CLARIFY outperforms existing methods in policy performance and labeling efficiency, generating high-quality embeddings that boost human feedback accuracy in real-world applications, making PbRL more practical.

## Acknowledgments

This work is supported by the Strategic Priority Research Program of the Chinese Academy of Sciences (No.XDA27040200), the NSFC (No. 62125304, 62192751, and 62073182), the Beijing Natural Science Foundation (L233005), the BNRist project (BNR2024TD03003) and the 111 International Collaboration Project (B25027). The author would like to express gratitude to Yao Luan for the constructive suggestions for this paper.

## Impact Statement

We believe that CLARIFY has the potential to significantly improve the alignment of reinforcement learning agents with human preferences, enabling more precise and adaptable AI systems. Such advancements could have broad societal implications, particularly in domains like healthcare, education, and autonomous systems, where understanding and responding to human intent is crucial. By enhancing the ability of AI to align with diverse human values and preferences, CLARIFY could promote greater trust in AI technologies, facilitate their adoption in high-stakes environments, and ensure that AI systems operate ethically and responsibly. These developments could contribute to the responsible advancement of AI technologies, improving their safety, fairness, and applicability in real-world scenarios.

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

# A. Additional Proofs and Theoretical Analysis

## A.1. Proof of Proposition 5.1: Margin Lower Bound

**Theorem A.1** (Proposition 5.1, restated). *Let $D_p$ be a distribution over trajectory-pairs $(\sigma_0, \sigma_1, p)$ where $p \in \{0, 1, \texttt{no\_cop}\}$. Define*

$$P = \{(\sigma_0, \sigma_1) \mid p \in \{0, 1\}\}, \qquad N = \{(\sigma_0, \sigma_1) \mid p = \texttt{no\_cop}\}. \tag{13}$$

*Let $\phi^* = \arg\min_{\phi} \mathcal{L}_{\mathrm{amb}}(\phi)$. Assume:*

1. *(Compactness) The support of $D_p$ in $(\sigma_0, \sigma_1)$-space is compact, and each $f_\phi$ is continuous.*

2. *(Richness) For any two pairs $(z_0^+, z_1^+)$ with $(\sigma_0^+, \sigma_1^+) \in P$ and $(z_0^-, z_1^-)$ with $(\sigma_0^-, \sigma_1^-) \in N$, there exists an infinitesimal perturbation of $\phi^*$ that simultaneously changes $\ell(z_0^+, z_1^+)$ and $\ell(z_0^-, z_1^-)$ without affect other embeddings.*

*Write $z_i = f_{\phi^*}(\sigma_i)$ and define*

$$d_{\mathrm{min}}^+ = \inf_{(\sigma_0, \sigma_1) \in P} \ell(z_0, z_1), \quad d_{\mathrm{max}}^- = \sup_{(\sigma_0, \sigma_1) \in N} \ell(z_0, z_1). \tag{14}$$

*Then there exists a constant $\delta > 0$ such that*

$$d_{\mathrm{min}}^+ \geq d_{\mathrm{max}}^- + \delta > 0. \tag{15}$$

*Proof.* We first show that

$$d_{\mathrm{min}}^+ > d_{\mathrm{max}}^-. \tag{16}$$

Set

$$A(\phi) := \mathbb{E}_{(\sigma_0, \sigma_1) \in P}\big[\ell\big(f_\phi(\sigma_0), f_\phi(\sigma_1)\big)\big], \quad B(\phi) := \mathbb{E}_{(\sigma_0, \sigma_1) \in D}\big[\ell\big(f_\phi(\sigma_0), f_\phi(\sigma_1)\big)\big], \tag{17}$$

so that $\mathcal{L}_{\mathrm{amb}}(\phi) = -A(\phi) + B(\phi)$. Since $\phi^*$ is a global minimizer,

$$\mathcal{L}_{\mathrm{amb}}(\phi^*) \leq \mathcal{L}_{\mathrm{amb}}(\phi^* + \Delta\phi) \implies -A(\phi^*) + B(\phi^*) \leq -A(\phi^* + \Delta\phi) + B(\phi^* + \Delta\phi) \tag{18}$$

for every infinitesimal admissible $\Delta\phi$.

Suppose for contradiction that

$$d_{\mathrm{min}}^+ \leq d_{\mathrm{max}}^-. \tag{19}$$

Then there exist one distinguished pair $(\sigma_0^+, \sigma_1^+)$ and one ambiguous pair $(\sigma_0^-, \sigma_1^-)$ such that

$$\ell\big(z_0^+, z_1^+\big) \leq \ell\big(z_0^-, z_1^-\big). \tag{20}$$

Because $\ell$ is strictly increasing in the Euclidean norm, we can choose a tiny $\varepsilon > 0$ and perturb

$$z_0^+ \mapsto z_0^+ + \tfrac{\varepsilon}{2} u^+, \quad z_1^+ \mapsto z_1^+ - \tfrac{\varepsilon}{2} u^+, \quad z_0^- \mapsto z_0^- - \tfrac{\varepsilon}{2} u^-, \quad z_1^- \mapsto z_1^- + \tfrac{\varepsilon}{2} u^-, \tag{21}$$

where $u^+$ is the unit-vector from $z_1^+$ to $z_0^+$ (and similarly $u^-$), so as to increase $\ell(z_0^+, z_1^+)$ by some $\delta' > 0$ and decrease $\ell(z_0^-, z_1^-)$ by some $\delta'' > 0$. By the richness assumption, this arises from some $\phi^* + \Delta\phi$.

Under this perturbation,

$$A(\phi^* + \Delta\phi) = A(\phi^*) + \tfrac{1}{N^+} \delta' + \text{(higher-order terms)}, \quad B(\phi^* + \Delta\phi) = B(\phi^*) - \tfrac{1}{N^-} \delta'' + \cdots, \tag{22}$$

so

$$\mathcal{L}_{\mathrm{amb}}(\phi^* + \Delta\phi) - \mathcal{L}_{\mathrm{amb}}(\phi^*) = \underbrace{-\tfrac{1}{N^+} \delta'}_{<0} + \underbrace{\tfrac{1}{N^-} \delta''}_{>0} < 0 \tag{23}$$

for sufficiently small $\varepsilon$. This contradicts the global optimality of $\phi^*$. Therefore

$$d_{\mathrm{min}}^+ > d_{\mathrm{max}}^-, \tag{24}$$

as claimed.

Then we show the strict gap via compactness. Since the support of $D_p$ is compact and $f_{\phi^*}$, $\ell$ are continuous, the image sets

$$S^+ = \{\ell(z_0, z_1) \mid (\sigma_0, \sigma_1) \in P\}, \quad S^- = \{\ell(z_0, z_1) \mid (\sigma_0, \sigma_1) \in N\} \tag{25}$$

are compact and, by the separation result, disjoint in $\mathbb{R}$. Two disjoint compact subsets of $\mathbb{R}$ have a strictly positive gap:

$$\delta = \inf S^+ - \sup S^- > 0. \tag{26}$$

Therefore

$$d_{\min}^+ = \inf S^+ \geq \sup S^- + \delta = d_{\max}^- + \delta, \tag{27}$$

completing the proof. $\qquad\square$

### A.2. Proof of Proposition 5.2: Convex Separability

**Proposition A.2** (Margin Guarantee for $\mathcal{L}_{\mathrm{quad}}$). *Let $C^+$ and $C^- \subset \mathbb{R}^d$ be convex, and write*

$$\mu^+ = \mathbb{E}[z^+], \quad \mu^- = \mathbb{E}[z^-], \qquad w = \mu^+ - \mu^-, \quad b = -\tfrac{1}{2}\left(\|\mu^+\|^2 - \|\mu^-\|^2\right). \tag{28}$$

*Define the signed distance $\widetilde{d}(z) = (w^\top z + b)/\|w\|$. If $\phi$ globally minimizes*

$$\mathcal{L}_{\mathrm{quad}} = -\mathbb{E}\left[\ell(z^+, z_-') + \ell(z_+', z^-) - \ell(z^+, z^-) - \ell(z_+', z_-')\right], \tag{29}$$

*then for every $z^+ \in C^+$ and $z^- \in C^-$,*

$$\widetilde{d}(z^+) \geq \eta, \quad \widetilde{d}(z^-) \leq -\eta, \tag{30}$$

*where $\eta = \tfrac{1}{2}\|\mu^+ - \mu^-\|$.*

*Proof.* Writing out the expectation and differentiating under the integral sign, we can find

$$\frac{\partial \mathcal{L}_{\mathrm{quad}}}{\partial w} = 2\left(w - (\mu^+ - \mu^-)\right) = 0, \qquad \frac{\partial \mathcal{L}_{\mathrm{quad}}}{\partial b} = 2\left(b + \tfrac{1}{2}(\|\mu^+\|^2 - \|\mu^-\|^2)\right) = 0. \tag{31}$$

Hence, at any global minimum,

$$w = \mu^+ - \mu^-, \quad b = -\tfrac{1}{2}\left(\|\mu^+\|^2 - \|\mu^-\|^2\right). \tag{32}$$

A direct expansion shows

$$w^\top z + b = \tfrac{1}{2}\left(\|z - \mu^-\|^2 - \|z - \mu^+\|^2\right), \tag{33}$$

so

$$\widetilde{d}(\mu^+) = \frac{\|\mu^+ - \mu^-\|^2}{2\|\mu^+ - \mu^-\|} = \frac{1}{2}\|\mu^+ - \mu^-\| = \eta, \quad \widetilde{d}(\mu^-) = -\eta. \tag{34}$$

Since $\widetilde{d}(z)$ is an affine function of $z$ and $C^+$, $C^-$ are convex, the entire set $C^+$ must lie on or beyond the level set $\{\widetilde{d} = \eta\}$, and $C^-$ on or beyond $\{\widetilde{d} = -\eta\}$. Equivalently,

$$\widetilde{d}(z^+) \geq \eta, \quad \widetilde{d}(z^-) \leq -\eta, \tag{35}$$

for every $z^+ \in C^+$, $z^- \in C^-$, as claimed. $\qquad\square$

### A.3. Gradient Dynamics of $\mathcal{L}_{\mathbf{quad}}$ and $\mathcal{L}_{\mathbf{amb}}$

The gradient expressions reveal two key mechanisms that govern the dynamics of the embedding:

- **Parametric Alignment ($\mathcal{L}_{\mathbf{quad}}$):** The gradient $\nabla z^+ \mathcal{L}_{\mathrm{quad}} = 2(z^{+\prime} - z^{-\prime})$ induces a *relational drift*, which does not move $z^+$ towards fixed coordinates but instead navigates based on the *relative positions* of its augmented counterpart $z^{+\prime}$ and the negative sample $z^{-\prime}$. This leads to a self-supervised clustering effect, where trajectories with similar preference labels naturally group together in tight neighborhoods.

- **Discriminant Scaling ($\mathcal{L}_{\mathbf{amb}}$):** The distance-maximization gradient for distinguishable pairs, $-2(z_0 - z_1)$, enforces *repulsive dynamics*, similar to the Coulomb forces between same-charge particles. In contrast, the gradient for indistinguishable pairs, $2(z_0 - z_1)$, acts like spring forces, pulling uncertain samples toward neutral regions. This dynamic equilibrium prevents the embedding from collapsing while maintaining geometric coherence with the preference signals.

Together, these dynamics resemble a *potential field* in physics: high-preference trajectories act as attractors, low-preference ones as repulsors, and ambiguous regions as saddle points. This resulting geometry directly facilitates our query selection objective.

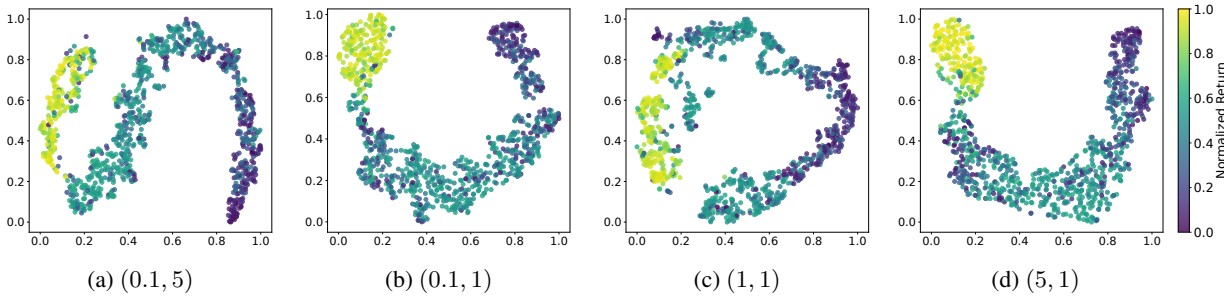

|          |          |          |          |
|:--------:|:--------:|:--------:|:--------:|
| (a) $(0.1, 5)$ | (b) $(0.1, 1)$ | (c) $(1, 1)$ | (d) $(5, 1)$ |

Figure 6: Embedding visualizations of the drawer-open task with $\epsilon = 0.5$, under different hyperparameters settings. The subfigure captions of are the values of $(\lambda_{\mathrm{amb}}, \lambda_{\mathrm{quad}})$, and $(\lambda_{\mathrm{amb}}, \lambda_{\mathrm{quad}}) = (0.1, 1)$ is the setting we use in experiments.

## B. Additional Experimental Results

**Ablation on hyperparameters.** As Figure 6 visualizes, when varying the values of $\lambda_{\mathrm{amb}}$, $\lambda_{\mathrm{quad}}$, and $\lambda_{\mathrm{norm}}$, the quality of embedding space remains largely unaffected. This demonstrates the robustness of CLARIFY to hyperparameter changes, as small adjustments do not significantly alter the quality of learned embeddings.

## C. Experimental Details

### C.1. Tasks

The robotic manipulation tasks from Metaworld (Yu et al., 2020) and the locomotion tasks from DMControl (Tassa et al., 2018) used in our experiments are shown in Figure 7. The descriptions of these tasks are as follows.

**Metaworld Tasks:**

1. Box-close: An agent controls a robotic arm to close a box lid from an open position.

2. Dial-turn: An agent controls a robotic arm to rotate a dial to a target angle.

3. Drawer-open: An agent controls a robotic arm to grasp and pull open a drawer.

4. Handle-pull-side: An agent controls a robotic arm to pull a handle sideways to a target position.

5. Hammer: An agent controls a robotic arm to pick up a hammer and use it to drive a nail into a surface.

**Algorithm 1** The proposed offline PbRL method using CLARIFY embedding

---

**Require:** Offline dataset $D$ with no reward signal, total feedback number $N_{\text{total}}$, query batch size $M$, number of discretization $n_{\text{bin}}$, number of gradient steps $n_{\text{init}}, n_{\text{emb}}, n_{\text{reward}}$, coefficients $\lambda_{\text{amb}}, \lambda_{\text{quad}}, \lambda_{\text{norm}}$
  *// Reward learning*
1: Initialize preference dataset $D_p = \emptyset$
2: Randomly select $M$ queries $\{(\sigma_0, \sigma_1, p)\}_M$, update $D_p \leftarrow \{(\sigma_0, \sigma_1, p)\}_M \cup D_p$
3: Optimize the encoder $f_\phi$ based on Eq. 11 by $n_{\text{init}}$ gradient steps
4: Optimize the reward model $\hat{r}_\theta$ based on Eq. 2 by $n_{\text{reward}}$ gradient steps
5: **while** total feedback $< N_{\text{total}}$ **do**
6:   Select $M$ queries $\{(\sigma_0, \sigma_1, p)\}_M$ based on the query selection method in Section 4.2, update $D_p \leftarrow \{(\sigma_0, \sigma_1, p)\}_M \cup D_p$
7:   Optimize the encoder $f_\phi$ based on Eq. 11 by $n_{\text{emb}}$ gradient steps
8:   Optimize the reward model $\hat{r}_\psi$ based on Eq. 2 by $n_{\text{reward}}$ gradient steps
9: **end while**
  *// Policy learning*
10: Label the offline dataset $D$ using reward model $\hat{r}_\theta$
11: Train the IQL policy $\pi_\theta$ using the relabeled dataset $D$

---

6. Peg-insert-side: An agent controls a robotic arm to insert a peg into a hole from the side.

7. Sweep-into: An agent controls a robotic arm to sweep an object into a target area.

**DMControl Tasks:**

1. Cheetah-run: A planar biped is trained to control its body and run on the ground.

2. Walker-walk: A planar walker is trained to control its body and walk on the ground.

We use the offline dataset from LiRE (Choi et al., 2024) for our experiments. LiRE allows for control over dataset quality by adjusting the size of the replay buffer (replay buffer size = data quality value * 100000), which provides different levels of dataset quality. The dataset quality in our experiments differs from the one used in LiRE, as detailed in Table 7. The number of total preference feedback used in our experiments is detailed in Table 8.

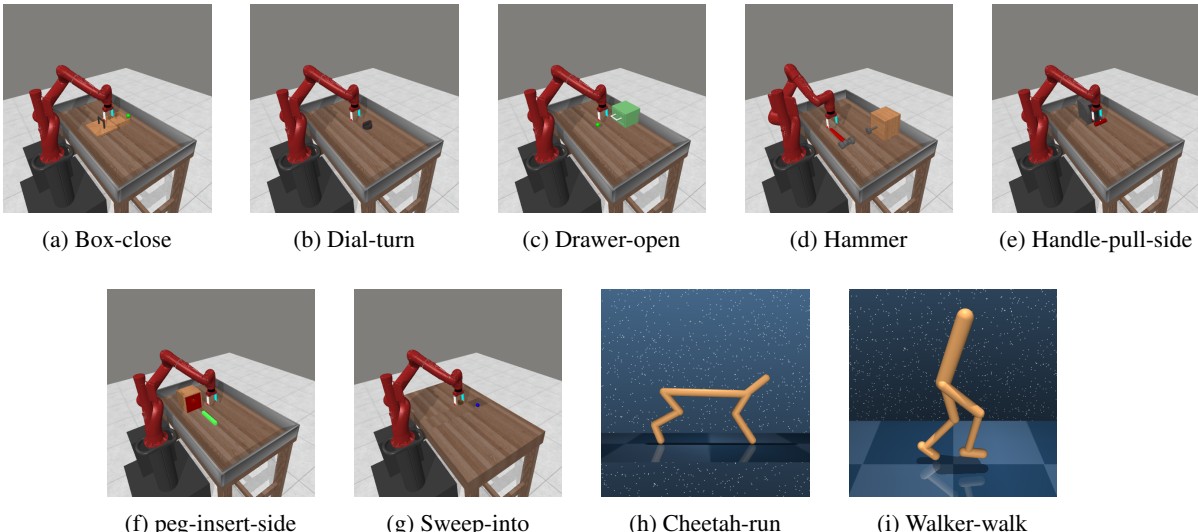

(a) Box-close   (b) Dial-turn   (c) Drawer-open   (d) Hammer   (e) Handle-pull-side

(f) peg-insert-side   (g) Sweep-into   (h) Cheetah-run   (i) Walker-walk

Figure 7: Nine tasks from Metaworld (a-g) and DMControl (h, i).

Table 7: The dataset quality in our experiments. Specifically, to prevent all the methods from failing under our teacher with the skip mechanism, we enhanced the dataset quality for several tasks.

| Task | Value of LiRE | Value of CLARIFY |
|---|---|---|
| Box-close | 8.0 | 9.0 |
| Dial-turn | 3.5 | 3.5 |
| Drawer-open | 1.0 | 1.0 |
| Hammer | 5.0 | 5.0 |
| Handle-pull-side | 1.0 | 2.5 |
| Peg-insert-side | 5.0 | 5.0 |
| Sweep-into | 1.0 | 1.5 |
| Cheetah-run | / | 6.0 |
| Walker-walk | 1.0 | 1.0 |

Table 8: Total feedback number $N_{\text{total}}$.

| Task | Value |
|---|---|
| Metaworld tasks | 1000 |
| Cheetah-run | 500 |
| Walker-walk | 200 |

### C.2. Implementation Details

In our experiments, MR, PT, OPRL, LiRE, and CLARIFY are all two-step PbRL methods. In these methods, the reward model is first trained, followed by offline RL using the trained reward model. The reward models used in CLARIFY, MR, and OPRL share the same structure, as outlined in Table 9. We use the trained reward model to estimate the reward for every $(s, a)$ pair in the offline RL dataset, and we apply min-max normalization to the reward values so that the minimum and maximum values are mapped to 0 and 1, respectively. We use IQL as the default offline RL algorithm. The total number of gradient descent steps in offline RL is 200,000, and we evaluate the success rate or episodic return for 20 episodes every 5,000 steps. For all baselines and our method, we run 6 different seeds. We report the average success rate or episodic return of the last five trained policies. The hyperparameters for offline policy learning are provided in Table 9.

We follow the official implementations of MR and PT[1], OPPO[2], and LiRE[3]. Note that LiRE treats queries with a too-small reward difference as equally preferred ($p = 0.5$), while in our setting, these queries are labeled as no_cop and excluded from reward learning.

For the BDT implementation in CLARIFY, we follow the implementation of HIM (Furuta et al., 2021a)[4], using the BERT architecture as the encoder and the GPT architecture as the decoder.

The code repository of our method is:

https://github.com/MoonOutCloudBack/CLARIFY_PbRL

The hyperparameters for both the baselines and our method are listed in Table 10.

### C.3. Details of the Intuitive Example

To validate the effectiveness of proposed $\mathcal{L}_{\text{quad}}$, we generated 1,000 data points with values uniformly distributed between 0 and 1. Each data point's embedding was initialized as a 2-dimensional vector sampled from a standard normal distribution. To simulate the issue of ambiguous queries, we defined queries with value differences smaller than 0.3 as ambiguous. We

---

[1]https://github.com/csmile-1006/PreferenceTransformer
[2]https://github.com/bkkgbkjb/OPPO
[3]https://github.com/chwoong/LiRE
[4]https://github.com/frt03/generalized_dt

then optimized these embeddings using the quadrilateral loss. To ensure stability during training, we imposed a penalty to constrain the L2 norm of the embedding vectors to 1. The learning rate was set to 0.1, and the hyperparameters $\lambda_{\text{quad}}$ and $\lambda_{\text{norm}}$ were set to 1 and 0.1, respectively.

Table 9: Hyperparameters of reward learning and policy learning.

|  | Hyperparameter | Value |
|---|---|---|
| | Optimizer | Adam |
| | Learning rate | 3e-4 |
| | Segment length $H$ | 50 |
| | Batch size | 128 |
| | Hidden layer dim | 256 |
| Reward model | Hidden layers | 3 |
| | Activation function | ReLU |
| | Final activation | Tanh |
| | # of updates $n_{\text{reward}}$ | 50 |
| | # of ensembles | 3 |
| | Reward from the ensemble models | Average |
| | Query batch size $M$ | 50 |
| | Optimizer | Adam |
| | Critic, Actor, Value hidden dim | 256 |
| | Critic, Actor, Value hidden layers | 2 |
| | Critic, Actor, Value activation function | ReLU |
| IQL | Critic, Actor, Value learning rate | 0.5 |
| | Mini-batch size | 256 |
| | Discount factor | 0.99 |
| | $\beta$ | 3.0 |
| | $\tau$ | 0.7 |

## D. Human Experiments

**Preference collection.** We collect feedback from human labelers (the authors) familiar with the tasks. Specifically, they watched video renderings of each segment and selected the one that better achieved the objective. Each trajectory segment was 1.5 seconds long, consisting of 50 timesteps. For Figure 5, labelers provided labels for 20 queries for each reward difference across 3 random seeds. For Figure 4, we ran 3 random seeds for each method, with labelers providing 20 preference labels for each run. For Table 5, we ran 3 random seeds for each method, with labelers providing 100 preference labels for each run, and the preference batch size $M = 20$.

**Prompts given to human labelers.** The prompts below describe the task objectives and guide preference judgments.

**Metaworld dial-turn task.**

Task Purpose:

In this task, you will be comparing two segments of a robotic arm trying to turn a dial. Your goal is to evaluate which segment performs better in achieving the task's objectives.

Instructions:

- Step 1: First, choose the segment where the robot's arm reaches the dial more accurately (the reach component).

- Step 2: If the reach performance is the same in both segments, then choose the one where the robot's gripper is closed more appropriately (the gripper closed component).

- Step 3: If both reach and gripper closure are equal, choose the segment that has the robot's arm placed closer to the target position (the in-place component).

**Metaworld hammer task.**

Task Purpose:

In this task, you will be comparing two segments where a robotic arm is hammering a nail. The aim is to evaluate which segment results in a better execution of the hammering process.

Instructions:

- Step 1: First, choose the segment where the hammerhead is in a better position and the nail is properly hit (the in-place component).

- Step 2: If the hammerhead positioning is similar in both segments, choose the one where the robot is better holding the hammer and the nail (the grab component).

- Step 3: If both the hammerhead position and grasping are the same, select the segment where the orientation of the hammer is more suitable (the quaternion component).

**DMControl walker-walk task.**

Task Purpose:

In this task, you will compare two segments where a bipedal robot is attempting to walk. Your goal is to determine which segment shows better walking performance.

Instructions:

- Step 1: First, choose the segment where the robot stands more stably (the standing reward).

- Step 2: If both segments have the same stability, choose the one where the robot moves faster or more smoothly (the move reward).

- Step 3: If both standing and moving are comparable, select the segment where the robot maintains a better upright posture (the upright reward).

## E. Introduction to Bi-directional Decision Transformer

**Introduction to Hindsight Information Matching (HIM).** Hindsight Information Matching (HIM) (Furuta et al., 2021a) aims to improve offline reinforcement learning by aligning the state distribution of learned policies with expert demonstrations. Instead of directly imitating actions, HIM minimizes the discrepancy between the marginal state distributions of the learned and expert policies, ensuring that the agent visits states similar to those in expert trajectories. This approach is particularly effective for multi-task and imitation learning settings, where aligning state distributions across different tasks can enhance generalization.

**The main idea of BDT.** BDT is designed for offline one-shot imitation learning, also known as offline multi-task imitation learning. In this setting, the agent must generalize to unseen tasks after observing only a few expert demonstrations. Standard Decision Transformer (DT) faces challenges in such scenarios due to its reliance on autoregressive action prediction and task-specific fine-tuning. In contrast, BDT improves generalization by utilizing bidirectional context, enabling it to infer useful patterns from limited data. This makes BDT particularly effective for imitation learning tasks where the agent needs to efficiently adapt to new environments with just a small number of demonstrations.

BDT extends the Decision Transformer (DT) framework by incorporating bidirectional context, which improves sequence modeling in offline reinforcement learning. Unlike standard DT, which predicts actions autoregressively using causal attention, BDT adds an anti-causal transformer that processes the trajectory in reverse order. This anti-causal component enables BDT to use both past and future information, making it highly effective for state-marginal matching and imitation learning.

**The architecture of BDT.** In this paper, we utilize the encoder-decoder framework of BDT to learn the trajectory representation. The architecture of BDT consists of two transformer networks: a causal transformer that models forward dependencies in the trajectory, and an anti-causal transformer that captures backward dependencies. This bidirectional structure allows BDT to extract richer temporal patterns, improving its ability to learn from diverse offline datasets.

Table 10: Hyperparameters of baselines and CLARIFY.

|  | Hyperparameter | Value |
|---|---|---|
| OPRL | Initial preference labels | $M = 50$ |
|  | Query selection | Maximizing disagreement |
| PT | Optimizer | AdamW |
|  | # of layers | 1 |
|  | # of attention heads | 4 |
|  | Embedding dimension | 256 |
|  | Dropout rate | 0.1 |
| OPPO | Optimizer | AdamW |
|  | Learning rate | 1e-4 |
|  | $z^*$ learning rate | 1e-3 |
|  | Number of layers 4 |  |
|  | Number of attention heads | 4 |
|  | Activation function | ReLU |
|  | Triplet loss margin | 1 |
|  | Batch size | 256 |
|  | Context length | 50 |
|  | Embedding dimension | 16 |
|  | Dropout | 0.1 |
|  | Grad norm clip | 0.25 |
|  | Weight decay | 1e-4 |
|  | $\alpha$ | 0.5 |
|  | $\beta$ | 0.1 |
| LiRE | Reward model | Linear |
|  | RLT feedback limit $Q$ | 100 |
| CLARIFY | Embedding dimension | 16 |
|  | Activation function | ReLU |
|  | $\lambda_{\text{amb}}$ | 0.1 |
|  | $\lambda_{\text{quad}}$ | 1 |
|  | $\lambda_{\text{norm}}$ | 0.1 |
|  | Batch size | 256 |
|  | Context length | 50 |
|  | Dropout | 0.1 |
|  | Number of layers 4 |  |
|  | Number of attention heads | 4 |
|  | Grad norm clip | 0.25 |
|  | Weight decay | 1e-4 |
|  | $n_{\text{init}}$ | 20000 |
|  | $n_{\text{emb}}$ | 2000 |

