# OpenReview forum: "CLARIFY: Contrastive Preference Reinforcement Learning for Untangling Ambiguous Queries"
_ICML.cc/2025/Conference — ICML 2025 poster_

### Official Review · Reviewer_985K · 2025-03-12

**Overall Recommendation:** 2

**Summary:**

This paper address the query selection problem of PbRL. The authors propose a representation learning algorithm that embeds trajectories into high dimensional vectors and enlarge the distance between unambiguous trajectories pairs. The authors compare their proposed method with existing PbRL methods.

**Claims And Evidence:**

The experiment results include a table for task performance and an ablation study shows that their method can increase the clarity of query. These results confirm the efficacy of their method.

In the meantime, this paper does not explicitly mention if the preference labels are collected in a single round or multiple rounds. If the preference labels are collected in multiple rounds, then the embedding space at round $T$ depends on previous rounds. In this setting, the method proposed in this paper can be considered as a exploitation strategy. In the meantime, exploration is also important: it seems that allocating label budget in ambiguous queries could be beneficial, as it might help us resolve the decision boundary.

If the preference labels are collected in multiple rounds, could you please results for label efficiency? In other words, how performance changes as we get more preference labels?

**Essential References Not Discussed:**

No.

**Experimental Designs Or Analyses:**

Yes, I have checked the experiment results.

1. This paper overlooks the analysis of exploitation and exploration tradeoff in the experiments.
2. Since the author claims that their method mitigates overfitting issue, please provide results for the quality of reward learning, i.e. accuracy on test preference set.

**Methods And Evaluation Criteria:**

This paper compares methods using the task performance of offline RL, which is standard in this area.

**Other Comments Or Suggestions:**

1. In line 199-200, I think the "positive" samples should be $\sigma_+$ and $\sigma'_+$.

**Other Strengths And Weaknesses:**

Ambiguous queries do not have labels of "1" or "-1", so they are not included in the samples used for minimizing eq. 6. The paper lacks explanation for why minimizing eq 6 resolves representation collapse of ambiguous queries.

**Questions For Authors:**

1. I notice that the preference loss $\mathcal{L}_\text{reward}$ does not appear in e.q. (11). How do you get the reward function from the trajectory embeddings? What is the reason of not learning the representations with both representation losses and reward learning losses?

2. Are the preference labels collected in single round or multiple rounds? Could you please provide results for the exploitation-exploration trade-off?

3. Can you provide results for reward fitting (i.e. accuracy on unseen preference samples)?

**Relation To Broader Scientific Literature:**

Embedding entire trajectories into a high-dimensional space and adjust the embeddings based on the proximity between trajectories is an interesting idea for the offline RL and the RL literature.

**Theoretical Claims:**

No, I did not check the correctness of the proofs.

---

> ### Author Rebuttal · Authors · 2025-03-30
>
> Dear Reviewer,
>
>
> Thanks for your valuable and detailed comments.
> We hope the following statement clear your concern.
> **We conducted additional experiments and the results are shown in the [link](https://docs.google.com/document/d/e/2PACX-1vS0ZIKigh-syAaNtcr2Udzk8katEE6AtC0OA23Xveb1dUqzFtMws64U6o6GFUep_BTzQ0EaA770n88P/pub).**
>
>
>
> **Claim 1, Experimental Designs 1 and Q2: Preference collection and exploration-exploitation trade-off.**
>
> **A for Claim 1, Experimental Designs 1 and Q2:**
> The preference labels in our study are collected over multiple rounds, with the embedding space updated iteratively as new queries are selected.
> As for the exploration-exploitation trade-off, CLARIFY prioritizes exploitation by focusing on unambiguous queries, while its reject sampling strategy (Sec. 4.2) inherently supports exploration. This strategy diversifies the queries by sampling from a distribution, avoiding overemphasizing the "clearest" pairs.
>
> As suggested, we conducted experiments to evaluate the exploration-exploitation trade-off.
> We compared CLARIFY against two baselines: (1) pure exploration (random query selection, referred to as "Random") and (2) pure exploitation (maximizing the density for clearly-distinguished queries $\rho_\text{clr}(d_\text{emb})$, referred to as "Exploitation").
> As shown in Tables 1 and 2 in the supplement link, CLARIFY outperforms Random by over 20\% in success rate, and surpasses Exploitation by over 15\%. These results demonstrate an effective exploration-exploitation trade-off.
>
>
>
> **Claim 2: Label efficiency.**
>
> **A for Claim 2:**
> As suggested, we evaluate CLARIFY's label efficiency from 50 to 2000. Table 3 in the supplement link shows that CLARIFY consistently outperforms baselines under various query numbers.
> In addition, we observed that the performance of CLARIFY with 100 queries approaches that of MR with 1000 queries, demonstrating CLARIFY's effectiveness in achieving high performance with fewer labels.
>
>
>
> **Experimental Designs 2 and Q3: Reward fitting accuracy on the test set.**
>
> **A for Experimental Designs 2 and Q3:**
> As suggested, we evaluate CLARIFY's reward fitting accuracy on the test set. Table 4 in the supplement link reflects the effectiveness of reward fitting, showing that CLARIFY attains about 3\% higher accuracy than OPRL on most tasks.
>
>
>
> **W1: How Eq. 6 resolves representation collapse.**
>
> **A for W1:**
> The quadrilateral loss $\mathcal L_\text{quad}$ prevents representation collapse by enforcing preference-aware geometry:
>
> 1. For clear preferences, it creates a hyperplane separating good and bad trajectories (Proposition 5.2).
> 2. For ambiguous pairs, while $\mathcal L_\text{amb}$ minimizes their embedding distance, $\mathcal L_\text{quad}$'s contrastive gradient prevents trivial clustering of these pairs.
>
> To support this, we conduct an ablation study on $\mathcal L_\text{amb}$, as shown in Table 5 in the supplement link. Removing $\mathcal L_\text{quad}$ degrades performance by about 10\% on average, validating its necessity.
>
>
>
> **A for Other Comments:**
> Thank you for your keen attention to detail! We have corrected the positive and negative samples to $(\sigma_+, \sigma^\prime_+), (\sigma_-, \sigma^\prime_-)$ in the revised manuscript.
>
>
>
> **Q1: $\mathcal L_\text{reward}$ optimization.**
>
> **A for Q1:**
> To elaborate, we present an additional figure (Figure 1 in the supplement link) that illustrates the architecture of CLARIFY.
> As shown in the figure, the reward model is updated using the loss function $\mathcal L_\text{reward}$ (Eq. 2) during the reward learning phase, while the embedding space is updated according to Eq. 11 during the embedding learning phase.
> These two phases are strictly decoupled. Thus, the loss function $\mathcal L_\text{reward}$ is not incorporated into Eq. 11.
> Training the reward model solely with the Bradley-Terry loss is a standard approach in the PbRL literature [1,2,3].
> This separation allows preference learning to focus on human feedback, while embeddings specialize in distinguishing trajectory pairs.
>
>
>
> Thanks again for the valuable comments. We hope our response has cleared your concerns. We are looking forward to more discussions.
>
> [1] Lee, Kimin, et al. "B-pref: Benchmarking preference-based reinforcement learning." arXiv preprint arXiv:2111.03026 (2021).
>
> [2] Shin, Daniel, et al. "Benchmarks and Algorithms for Offline Preference-Based Reward Learning." Transactions on Machine Learning Research.
>
> [3] Cheng, Jie, et al. "RIME: Robust Preference-based Reinforcement Learning with Noisy Preferences." International Conference on Machine Learning. PMLR, 2024.

---

### Official Review · Reviewer_znbK · 2025-03-13

**Overall Recommendation:** 4

**Summary:**

This paper presents CLARIFY, a method that selects unambiguous queries that humans can more easily label. It does this by learning a meaningful embedding space using two contrastive losses. This allows for weaker teachers to provide meaningful feedback on the selected trajectories. Experimental results in continuous control tasks show that CLARIFY can significantly improve labelling efficiency and improve policy performance.

## After Rebuttal
The authors have nicely addresses my concerns. I will raise my score to a 4.

**Claims And Evidence:**

The only claim I have an issue with is their claim that Mu 2024 cannot be applied to offline settings (this claim is made on line 162). I have not looked into Mu 2024 deeply, but I do not see why it cannot be applied to offline settings with some simple modifications. Furthermore, I don't understand why the offline setting is important for the tasks the authors consider.

The main claims about the performance gains of Clarify are solid. They conduct many experiments and it seems like Clarify can indeed improve performance in most settings.

**Essential References Not Discussed:**

Mu 2024 is cited, but not discussed in depth. Since Mu 2024 aims to accomplish a very similar goal as CLARIFY (albeit in the online setting), I think CLARIFY's novelty and contribution in comparison to Mu 2024 should be discussed in more detail.

**Experimental Designs Or Analyses:**

Yes I did check the soundness of the experimental design. The experimental design and analysis is sound. They compare their algorithm with many baselines, and they conduct experiments in a wide variety of environments. In addition, they report five independent runs for each algorithm, and report standard deviation for all experiments. The main experimental results I am referring to are in Table 1.

They also show that CLARIFY results in queries that are more distinguishable to humans (Figure 4), and these results also seem sound.

**Methods And Evaluation Criteria:**

The experiments and evaluation criteria do make sense. This paper works to improve RLHF, and some of the earliest papers on RLHF focused on continuous control [1]. They evaluate based on the average ground-truth reward received, which makes sense. However their experiments are all conducted on relatively simple continuous control environments. One concern I have is that their approach of only selecting easy samples to train on will work better in easy settings than in hard settings. This means their experimental setup may overestimate the utility of their method.

[1] Christiano, Paul F., et al. "Deep reinforcement learning from human preferences." Advances in neural information processing systems 30 (2017).

**Other Comments Or Suggestions:**

none

**Other Strengths And Weaknesses:**

Strengths
- The writing is easy to understand, and the presentation is nice
- The approach seems to be novel
- The experiments are thorough and their approach shows solid improvement compared to baselines.

Weaknesses:
- The authors do compare in depth to Mu 2024 (why can’t it be included in the offline setting?)
- There is no discussion of offline versus online approaches. Why do the authors sample queries from the offline dataset, rather than sampling new on-policy queries from the current policy? It seems like this would be a more effective approach in all of the experimental settings the authors consider.

**Questions For Authors:**

See the above sections.

**Relation To Broader Scientific Literature:**

The authors approach seems novel and very relevant to data selection for RLHF, which is a popular area at the moment. I think the whole CLARIFY framework is novel, but it needs more detailed comparison to Mu 2024.

**Theoretical Claims:**

no, I did not check them for correctness.

---

> ### Author Rebuttal · Authors · 2025-03-30
>
> Dear Reviewer,
>
> Thanks for finding our paper nice presentation, novel, thorough experiments and solid improvement. We hope the following statement clear your concern.
>
> **Claim and W1: Comparison to [1] and its offline applicability.**
>
> **A for Claim and W1:**
> - While both CLARIFY and [1] attempt to tackle ambiguous queries, their approaches are fundamentally different.
> [1] address ambiguous queries by learning diverse skills through unsupervised exploration.
> The requirement for online exploration makes it incompatible with offline settings, where agents cannot interact with the environment.
> In contrast, CLARIFY operates entirely offline by learning preference-informed trajectory embeddings through contrastive learning.
> This eliminates the need for online exploration, making CLARIFY more suitable for real-world scenarios where environment interaction is costly, risky, or unavailable.
>
> - Additionally, CLARIFY optimizes query distinguishability directly via embedding distances, whereas [1] relies on skill diversity as a proxy.
> This independence from skill diversity enhances CLARIFY's robustness in tasks with constrained skill exploration (e.g., limited state-space traversability), which eliminates the reliance on costly exploration for discovering diverse skills.
>
> **Methods 1: Task complexity and generalizability.**
>
> **A for M1:**
> We appreciate the reviewer’s concern regarding task complexity.
> Our experiments include challenging tasks, such as MetaWorld's peg-insert-side, which require multi-step reasoning and are more complex than prior RLHF [2] evaluations focused on locomotion.
> CLARIFY significantly outperforms baselines across these tasks, as shown in Table 1 of the paper, demonstrating its effectiveness beyond simple settings.
>
> **W2: Importance of offline setting and query sampling.**
>
> **A for W2:**
> Offline reinforcement learning assumes agents learn solely from a fixed pre-collected dataset without environment interaction. It is critical for safety-sensitive domains (e.g., healthcare, autonomous systems), where real-time exploration is either unsafe or impractical.
> In such cases, agents must rely on pre-collected datasets for learning, and thus, queries should be sampled from this static dataset rather than generated on-policy.
> This aligns with CLARIFY's approach, which leverages offline datasets and contrastive learning to ensure the method's applicability in real-world, offline settings.
>
> We sincerely thank the reviewer again for the timely and valuable comments. We hope that our response and additional experimental results have cleared most of your concerns.
>
> [1] Mu, Ni, et al. "S-EPOA: Overcoming the Indistinguishability of Segments with Skill-Driven Preference-Based Reinforcement Learning." arXiv preprint arXiv:2408.12130 (2024).
>
> [2] Christiano, Paul F., et al. "Deep reinforcement learning from human preferences." Advances in neural information processing systems 30 (2017).

---

### Official Review · Reviewer_1xXL · 2025-03-13

**Overall Recommendation:** 3

**Summary:**

This paper proposes an offline PbRL framework, CLARIFY, to address challenges arising from ambiguous queries. The method learns a trajectory embedding space through contrastive learning and utilizes the learned embedding to maximize the selection of clearly distinguished queries via rejection sampling, improving human labeling efficiency and achieving state-of-the-art results in offline PbRL settings.

## Update After Rebuttal: I raised my score from 2 to 3 as the authors' rebuttal addressed most of my concerns.

**Claims And Evidence:**

The paper claims that the proposed embedding space is meaningful and coherent, as high-performance trajectories form one cluster, low-performance trajectories form another, and intermediate trajectories transition smoothly between them. The authors support this claim using Figures 2 and 3, which visualize the learned embedding spaces. My understanding is that Figure 2 represents the embedding space learned using only quadrilateral loss, while Figure 3 shows the embedding space learned with both ambiguity loss and quadrilateral loss. However, they appear similar. It would be helpful to highlight the failure points of the embedding space when using only ambiguity loss or only quadrilateral loss and demonstrate how combining these two losses addresses the problem.

**Essential References Not Discussed:**

There is potentially related literature that incorporates rejection sampling into PbRL or Preference Optimization, such as [1] Statistical Rejection Sampling Improves Preference Optimization, Liu et al., ICLR 2024.

**Experimental Designs Or Analyses:**

Adding a naive baseline that simply rules out ambiguous sample pairs with "p = no_cop" and comparing its performance with the proposed method would help readers better appreciate the effectiveness of the proposed approach.

Additionally, the proposed method introduces extra complexity to the PbRL algorithm by (1) learning an embedding space and (2) performing rejection sampling. Moreover, its final loss function involves several hyperparameters (e.g., $\lambda_{\mathrm{amb}}$, $\lambda_{\mathrm{quad}}$, $\lambda_{\mathrm{norm}}$). Comparing the computational costs and hyperparameter tuning efforts with other baselines would provide valuable insights into the true applicability of the proposed method.

**Methods And Evaluation Criteria:**

The proposed methods and benchmark datasets are reasonable for the problem or application at hand.

**Other Comments Or Suggestions:**

There is a typo in Line 179: 'amb' should be subscripted, e.g., $\mathcal{L}_{\mathrm{amb}}$.

**Other Strengths And Weaknesses:**

Strengths:

- The paper is well-written and easy to follow.
- The claims are largely supported by visualizations and theoretical analyses.
- The paper evaluates the method on diverse benchmark datasets, demonstrating its real-world applicability.

Weaknesses:

- The effect of each loss term in the final proposed loss is not empirically studied in depth.
- The computational overhead of learning the embedding and the effort required for hyperparameter tuning are not thoroughly analyzed.

**Questions For Authors:**

Q1. Why and how do ambiguous queries hinder the practical application of PbRL? Is this issue solely related to labeling efficiency, or does it also impact policy training within the PbRL framework?

Q2. Could you elaborate on the differences between S-EPOA: Overcoming the Indistinguishability of Segments with Skill-Driven Preference-Based Reinforcement Learning, Mu et al., 2024, and the proposed method beyond the distinction between online and offline settings?

Q3. How do the computational overhead and the effort required for hyperparameter tuning of the proposed method compare to those of other baselines?

Q4. Could you visualize the failure points of the embedding space or the performance results when using only ambiguity loss or only quadrilateral loss and demonstrate how combining these two losses resolves the issue?

Q5. How does the proposed method compare to a naive approach that simply rules out pairs with 'p = no_cop'?

**Relation To Broader Scientific Literature:**

Designing effective reward learning within the PbRL framework is closely related to the broader scientific literature.

**Theoretical Claims:**

The theoretical claims appear reasonable and effectively cover the core concept proposed in the paper.

---

> ### Author Rebuttal · Authors · 2025-03-30
>
> Dear Reviewer,
>
> Thanks for your valuable and detailed comments.
> We hope the following statement clear your concern.
> **We conducted additional experiments and the results are shown in the [link](https://docs.google.com/document/d/e/2PACX-1vQX0KIRCSWV8LrON718raf-d_BL75LRXMY5yB-Ts28kW0BZIVyWHan0kgw54vnZQtuxp1ODwe4IH1ws/pub).**
>
>
>
> **Claim, W1 and Q4: Embedding failure cases.**
>
> **A for Claim and Q4:**
> We would like to clarify that Figure 2(c) in the original paper visualizes embeddings trained with both losses rather than using only quadrilateral loss.
>
> As suggested, we visualize the failure modes in Figure 1 in the supplemental link.
> Specifically, using only $\mathcal L_\text{amb}$ causes erroneous clustering, where low and high return trajectory embeddings are intermixed (Figure 1(a) in the supplemental link).
> In contrast, using only $\mathcal L_\text{quad}$ yields densely packed embeddings with insufficient separation (Figure 1(b)).
> Combining both losses resolves these issues and leads to smooth, coherent clusters (Figure 1(c)).
> This illustrates their complementary roles in structuring the embedding space.
>
>
> **Experimental Design 1 and Q5: Naive baseline comparison.**
>
> **A for Experimental Design 1 and Q5:**
> As suggested, we compare CLARIFY with the naive approach that rules out pairs with `no_cop` preference labels, as shown in Table 1 in the supplement link.
> CLARIFY outperforms the naive approach (the "Naive" method in the table) by over 50\% on most tasks. This demonstrates the effectiveness of CLARIFY.
>
> **Experimental Design 2, W2 and Q3: Computational costs.**
>
> **A for Experimental Design 2, W2 and Q3:**
> - As suggested, we analyze the computational cost and the effort required for hyperparameter tuning of CLARIFY, as shown in Table 2 in the supplement link.
> CLARIFY incurs moderate computational overhead, roughly 2-3 times that of OPRL, primarily due to embedding learning.
> While training time may increase, CLARIFY effectively identifies more clearly distinguished queries, accelerating the labeling process.
> - On the other hand, CLARIFY demands minimal hyperparameter tuning.
> In the original paper, key hyperparameters ($\lambda_\text{amb},\lambda_\text{quad}$) were fixed across tasks with robust performance.
> To support this, we conduct additional experiments to visualize the embeddings under various hyperparameter configurations, as in Figure 2 in the supplement link, illustrating the stability of embeddings under parameter variations.
> We have added these results in the revised version.
>
> **Q1: Ambiguity's impact on PbRL.**
>
> **A for Q1:**
> Ambiguous queries hinder PbRL primarily in two ways:
> 1. Labeling efficiency: Human teachers often struggle to differentiate between similar segments, leading to skipped labels (`no_cop`) that waste annotation effort. As in Table 3 of the original paper, only about 50% of queries receive clear preference labels, indicating significant inefficiencies.
> 2. Reward learning accuracy: Labelers may produce random or incorrect preferences when segments are only marginally different. This can introduce errors to the reward model, ultimately degrading policy performance.
>
> **Q2: CLARIFY vs S-EPOA [1].**
>
> **A for Q2:**
> While both methods tackle ambiguous queries, they differ fundamentally in their approaches.
> S-EPOA addresses indistinguishability via unsupervised exploration and skill-based query selection in online settings.
> In contrast, CLARIFY utilizes contrastive learning to create trajectory embeddings informed by preferences, which enables offline query filtering.
> One of the key advantages of CLARIFY is that it eliminates the need for online exploration, making it particularly suitable for real-world applications, where interaction can be costly or risky.
> Furthermore, CLARIFY directly optimizes query distinguishability rather than relying on skill diversity as a surrogate.
> This independence from skill diversity enhances CLARIFY's robustness in tasks with constrained skill exploration (e.g., limited state-space traversability), eliminating reliance on costly exploration for discovering diverse skills.
>
> **Answer to Essential References and Other Comments:**
> Thank you for your keen attention to detail!
> We have subscripted $\mathcal L_\text{amb}$ throughout the manuscript. Additionally, we have discussed [2] in the related work section.
>
> Thanks again for the valuable comments. We sincerely hope our additional experimental results and explanation have cleared the concern.  More comments on further improving the presentation are also very much welcomed.
>
> [1] Mu, Ni, et al. "S-EPOA: Overcoming the Indistinguishability of Segments with Skill-Driven Preference-Based Reinforcement Learning." arXiv preprint arXiv:2408.12130 (2024).
>
> [2] Liu, Tianqi, et al. "Statistical Rejection Sampling Improves Preference Optimization." The Twelfth International Conference on Learning Representations.

---

> > ### Comment · Reviewer_1xXL · 2025-04-01
> >
> > Thank you for your thoughtful rebuttal. I appreciate the detailed responses to my questions and have increased my score to 3 in light of your clarifications.

---

> > > ### Author Response · Authors · 2025-04-02
> > >
> > > We would like to thank the reviewer for raising the score! We also appreciate the valuable comments, which helped us significantly improve the paper's strengths.

---

### Official Review · Reviewer_DiW3 · 2025-03-17

**Overall Recommendation:** 3

**Summary:**

This paper presents CLARIFY, an offline preference-based reinforcement learning (PbRL) algorithm, that leverages contrastive learning to organise the embedding space which is used to learn the reward function.

During the reward-learning phase, CLARIFY alternates between learning a reward via Bradley-Terry and a contrastive objective that encourages preferred state-action pairs to cluster together.

This paper additionally shows that under CLARIFY: i) the distance between two trajectory embeddings is has a lower-bound, and that ii) there is a hyperplane that separates the embeddings of all preferred trajectories, and all dispreferred trajectories.

Experiments show that Metaworld and DMControl tasks, where IQL policies using the reward learned by it CLARIFY outperforms in all but one scenario recent offline preference-based learning baselines (including OPRL, PT, OPPO and LiRE).

**Post-rebuttal update**

The main concern about this paper was whether CLARIFY was robust to noisier labellers, with additional ablations showing that it indeed was. There were also additional concerns regarding the clarity of the text and the data flow of the method that were equally clarified during rebuttal.

**Claims And Evidence:**

The following claims are made in the paper:

**C1. CLARIFY provides better performance than current offline PbRL baselines under non-ideal teachers**

This claim is mostly supported by the thorough experiments in Table 1, where CLARIFY beats the baseline in all tasks except Metaworld's `peg-insert-side`.

The robustness of CLARIFY to different non-ideal scripted teachers is not sufficiently investigated however. The teacher presented in CLARIFY can skip trajectories whose ground-truth reward is close, but will never flip preferences (ie choosing a non-preferred trajectory over a preferred one), or skip preferences where the ground-truth rewards are large.

Even with the presented scripted teacher, it is hard to gauge what is the effect of $\epsilon$ in the expected performance. How does CLARIFY perform if $\epsilon$ is ~0.3? What happens if $\epsilon$ is zero (ie no trajectory is labelled as `no_cop`)? It is not necessary for CLARIFY to outperform the baselines in these situations, but it is important to characterise its behaviour so that the community can understand when is best to use CLARIFY.

**C2. These improvements are both due to the space separation induced by CLARIFY, and the rejection sampling technique presented in the paper.**

Table 4, and 6 shows that best results are obtained when all elements of CLARIFY ($L_{amb}$, $L_{quad}$, and rejection sampling) are active. However, this analysis is only carried out for two meta-world tasks and with $\epsilon=0.5$. Other epsilon values and a few DMControl tasks should also be analysed.

In table 4, it would make sense to compare against an uncertainty-based sampling method like the one used in PEBBLE (Lee et al 2021a in the paper's bibliography).

Lastly, CLARIFY uses a Bidirectional Transformer, it is conceivable that the encoder of the transformer (which as far as I understand consumes the trajectories) is providing extra information to the policy. I would train implement the policy as a causal-decoder transformer only as an ablation to verify the model architecture is not behind the observed gains.

(It is possible that my assumptions of how CLARIFY is implemented are mistaken, this paper could benefit from an architecture diagram).

**Post-rebuttal update**

During rebuttal new experiments were added to address the concerns above. In particular, it became clear that CLARIFY is quite robust to $\epsilon$ changes and to other labellers (C.1). Similarly, experiments were added to show that the observed performance improvements were not due to the use of a bidirectional transformer (C.2)

**Essential References Not Discussed:**

None that I could find.

**Experimental Designs Or Analyses:**

All the analyses and ablations make sense, apart from the issues discussed above.

**Methods And Evaluation Criteria:**

Yes, the paper contains comparisons against recent baselines on commonly used tasks (MetaWorld and DMControl).

**Other Comments Or Suggestions:**

* What does `no_cop` stand for?

**Other Strengths And Weaknesses:**

**Other Strengths**:

* The paper includes a comparison against actual human labellers for the `walker-walk` task (though unfortunately its performance is approximately half of the performance achieved with a non-ideal teacher, cementing the need of analysis with other non-ideal teachers).

**Other Weaknesses**:

* Query clarity ratio is not very clearly defined in the manuscript. I believe it is 1 - (ratio of `no_cop`) labels provided by human labellers?
* Similarly, I am very confused by what is meant by clearly-distinguished queries in Table 3.
* The Impact Statement is well thought out, but it does not mention possible negative consequences of offline PbRL (namely the ability for laymen to very easily program an agent to carry out malicious tasks).

**Post-rebuttal update**
Authors addressed all the above issues. I would urge authors to adopt the definitions of `clearly-distinguished queries` and `clarity-ratio` present in the rebuttal.

**Questions For Authors:**

* Q1. When is equation (9) used in Algorithm 1? Is it simply a pre-training for the bi-directional encoder?
* Q2. The paper never really states what the distance metric $l$ in equation 5 is. I assume is simply the euclidean distance? Perhaps a more appropriate distance would be the cosine distance? In very high-dimensional spaces, it's very easy for two points to be very far apart.
* Q3. What is the number of queries used for Table 1? This should be clearly stated.
* Q4. How much do the results in Table 2 deteriorate under 50 queries? Does CLARIFY also ~77% of the IQL-with-ground-truth reward performance with only 100 samples for other tasks?
* Q5. The error bars on Figure 4 are quite large, particularly for `walker-walk`, could you run a statistical significance analysis to verify that CLARIFY and OPRL are indeed different?
* Q6. In Figure 5 what are the thresholds for Large Return Difference, Medium Return Difference, and Small Return Difference?
* Q7. Using T-SNE visualisation to prove that the embeddings are robust to changes in the loss $\lambda$ is very unconvincing. T-SNE has its own hyper-parameters and it may be ignoring important differences in the underlying space when projection to 2D.

**Post-rebuttal update**
Authors addressed all the above issues. I would urge authors to adopt the definitions of `clearly-distinguished queries` and `clarity-ratio` present in the rebuttal.

Regarding Q2, the research would likely benefit from further investigation of the effects of different distances (beyond a TSNE projection), but perhaps this could be left for further work.

**Relation To Broader Scientific Literature:**

CLARIFY is tackling the very challenging (and highly researched) problem of learning to solve a task without an explicit reward function from an offline dataset of interactions. In this context, the use of contrastive learning (which has been used elsewhere in Machine Learning to increase sample efficiency) is interesting.

**Theoretical Claims:**

I did not have time to go through the proofs of Propositions 5.1 and 5.2, since they are contained in the appendices. But the claims derived from these propositions seem sound.

I believe there is a typo in Proposition 5.2, $d(z^-, \mathcal{H})$ should be $\le \eta$ rather than $\le - \eta$.

---

> ### Author Rebuttal · Authors · 2025-03-30
>
> Dear Reviewer,
>
> Thanks for your valuable and detailed comments.
> **We conducted additional experiments and the results are shown in the [link](https://docs.google.com/document/d/e/2PACX-1vS7v9XEpXMFrH0skymO1RQUiXP2lcnnRoP114HpluBSSpvxE3vuRHNYJ1RwlggWB-rlihxrpdeVv53O/pub).**
>
> **C1.1: Robustness to non-ideal teachers.**
>
> **A for C1.1:**
> We evaluated CLARIFY with a new flipping teacher that randomly assigns preferences for close-reward queries. Table 1 shows that CLARIFY outperforms OPRL by over 20% in success rate, confirming its robustness to varied non-ideal feedback.
>
> **C1.2: CLARIFY's behavior across $\epsilon$ values.**
>
> **A for C1.2:** We evaluated CLARIFY with $\epsilon$ in 0$\sim$0.7. As shown in Table 2, at $\epsilon=0.3$, CLARIFY outperforms OPRL by 8~18\%. In contrast, at $\epsilon=0$, CLARIFY's query selection reverts to random sampling, leading to performance comparable to MR. This suggests that CLARIFY excels when distinguishing a query is difficult.
>
> **C2.1: Additional component analysis.**
>
> **A for C2.1:** We conduct component analysis on DMControl tasks with $\epsilon$=0.7. Tables 3 and 4 show that full CLARIFY ($\mathcal L_\text{amb}$ + $\mathcal L_\text{quad}$ + rejection sampling) consistently performs best, confirming component necessity.
>
> **C2.2: Comparison to PEBBLE.**
>
> **A for C2.2:** We compare CLARIFY to uncertainty-based sampling (PEBBLE). Table 5 shows that CLARIFY achieves better performance, demonstrating the effectiveness of our query selection strategy.
>
> **C2.3: Architecture impact.**
>
> **A for C2.3:** We illustrate our architecture in Figure 1, which shows that CLARIFY operates in two strictly decoupled phases. In the embedding training phase, the Bi-directional Decision Transformer encoder trains solely on trajectories using contrastive and reconstruction losses, without access to reward model or policy. In the reward learning phase, the reward model updates only on preference data without access to embedding information. This design prevents the encoder from leaking privileged information about future states or rewards.
>
>
> **Answer for Theoretical, Weaknesses, Other Comments:**
> - (**Theoretical**) Prop 5.2: corrected to $d(z^-,H)\ge\eta$.
> - (**Other Comments**) `no_cop` denotes cases where labelers consider queries too similar to specify a preference, resulting in a skipped label.
> - (**W1**) Query clarity ratio is defined as the proportion of clearly-distinguished queries to the total number of queries.
> - (**W2**) Clearly-distinguished queries are those where human preferences are clear (not `no_cop`).
> - (**W3**) Safety impact: A discussion of malicious use risks was added.
>
> **Q1, Q2, Q3, Q6: Clarity of the statement.**
>
> - **A for Q1:** Eq. 9 is integrated into the total training objective (Eq. 11) used in Algorithm 1 line 3. It is not a simple pretraining but trains the encoder continuously.
> - **A for Q2:** Distance metric $\ell$ is the Euclidean distance. We conduct additional experiments to compare the Euclidean and Cosine distances. Figure 2 shows that points in the embedding trained with Cosine distance cluster together, while Euclidean distance is more suitable for trajectory embedding.
> - **A for Q3:** Query number: 1000 for MetaWorld tasks, 500 for cheetah-run, 200 for walker-walk (Table 10).
> - **A for Q6:** The thresholds of Large, Medium, and Small in MetaWorld and DMControl tasks are 300/100/10 and 30/10/1 respectively.
>
> **Q4: Performance with fewer queries.**
>
> **A for Q4:** We evaluate CLARIFY with 50 to 2000 queries. Table 6 shows a slight performance decline for CLARIFY with only 50 queries, though it still outperforms MR. Table 7 shows CLARIFY's performance with 100 queries on various tasks, which reaches about 70\% of IQL's performance with ground truth rewards.
>
> **Q5: Statistical validation for walker-walk.**
>
> **A for Q5:** We conduct a statistical significance analysis for CLARIFY and OPRL.
> - Table 8 shows the 95% confidence intervals (CIs) of the query clarity ratio and label accuracy. The narrow intervals validate CLARIFY’s performance. Though walker-walk shows overlapping CI due to high environment stochasticity, CLARIFY's directional improvements in both metrics demonstrate the effectiveness of its query selection.
> - Additionally, we conducted independent two-sample t-tests comparing CLARIFY with OPRL. The experimental results in Table 9 show that CLARIFY achieves statistically significant improvements ($p<$0.05) in 5/6 tasks. These results confirm the advantage of CLARIFY's query selection over OPRL.
>
> **Q7: T-SNE visualization robustness.**
>
> **A for Q7:**
> Thanks for pointing out this issue, and we conduct visualizations across multiple random seeds (Figure 3), revealing consistent clustering patterns, indicating robust embeddings. Additionally, we provide PCA (Figure 4) visualizations to support our conclusion.
>
> We hope that our response has cleared most of your concerns.

---

> > ### Comment · Reviewer_DiW3 · 2025-04-07
> >
> > Thank you for your detailed rebuttal and for all the additional and thorough experiments. I would also urge the authors to include Figure 1 in the manuscript.
> >
> > Based on the authors responses, I have raised my review score to a 3.

---

> > > ### Author Response · Authors · 2025-04-07
> > >
> > > We would like to thank the reviewer for raising the score! We also appreciate the valuable comments, which helped us significantly improve the paper's strengths.

---

### Decision · Program_Chairs · 2025-05-01

**Decision:**

Accept (poster)

**Comment:**

This work introduces CLARIFY, an offline PBRL method that incorporates contrastive learning for reward learning from human preferences. The authors demonstrate that: i) contrastive learning ensures a lower bound on the distance between two trajectory embeddings, and ii) a hyperplane can effectively separate the embeddings of preferred and dispreferred trajectories. Experimental results on MetaWorld and DMControl tasks demonstrate the utility of the proposed method.

Some reviewers raised concerns (e.g., robustness against noisier labelers and comparisons with Mu 2024), but most of these concerns were addressed during the rebuttal phase. I believe that accepting this work would make a valuable contribution to the community. Therefore, I recommend a weak accept.